# CellSeg3D, Self-supervised 3D cell segmentation for fluorescence microscopy

Cyril Achard[1], Timokleia Kousi[1], Markus Frey[1], Maxime Vidal[1], Yves Paychere[1], Colin Hofmann[1], Asim Iqbal[1], Sebastien B Hausmann[1], Stéphane Pagès[2], Mackenzie Weygandt Mathis[1]*

[1]Brain Mind Institute and Neuro X, École Polytechnique Fédérale de Lausanne (EPFL), Geneva, Switzerland; [2]Wyss Center for Bio and Neuroengineering, Geneva, Switzerland

## eLife Assessment

This **important** work presents a self-supervised method for the segmentation of 3D cells in fluorescent microscopy images, conveniently packaged as a Napari plugin and tested on an annotated dataset. The segmentation method is **solid** and compares favorably to other learning-based methods and Otsu thresholding on four datasets, offering the possibility of eliminating time-consuming data labeling to speed up quantitative analysis. This work will be of interest to a wide variety of laboratories analysing fluorescently labeled images.

**Abstract** Understanding the complex three-dimensional structure of cells is crucial across many disciplines in biology and especially in neuroscience. Here, we introduce a set of models including a 3D transformer (SwinUNetR) and a novel 3D self-supervised learning method (WNet3D) designed to address the inherent complexity of generating 3D ground truth data and quantifying nuclei in 3D volumes. We developed a Python package called CellSeg3D that provides access to these models in Jupyter Notebooks and in a napari GUI plugin. Recognizing the scarcity of high-quality 3D ground truth data, we created a fully human-annotated mesoSPIM dataset to advance evaluation and benchmarking in the field. To assess model performance, we benchmarked our approach across four diverse datasets: the newly developed mesoSPIM dataset, a 3D platynereis-ISH-Nuclei confocal dataset, a separate 3D Platynereis-Nuclei light-sheet dataset, and a challenging and densely packed Mouse-Skull-Nuclei confocal dataset. We demonstrate that our self-supervised model, WNet3D – trained without any ground truth labels – achieves performance on par with state-of-the-art supervised methods, paving the way for broader applications in label-scarce biological contexts.

*For correspondence:
mackenzie.mathis@epfl.ch

Competing interest: The authors declare that no competing interests exist.

## Introduction

Recent advancements in three-dimensional (3D) imaging techniques have provided unprecedented insights into cellular and tissue-level processes. In addition to confocal imaging and other fluorescent techniques, imaging systems based on light-sheet microscopy (LSM), such as the mesoscopic selective plane-illumination microscopy (mesoSPIM) initiative (*Voigt et al., 2019*), have emerged as powerful tools for high-resolution 3D imaging of biological specimens. Due to its minimal phototoxicity and ability to capture high-resolution 3D images of thick biological samples, it has been a powerful new approach for imaging thick samples, such as the whole mouse brain, without the need for sectioning.

The analysis of such large-scale 3D datasets presents a significant challenge due to the size, complexity, and heterogeneity of the samples. Yet, accurate and efficient segmentation of cells is a crucial step towards density estimates as well as quantification of morphological features. To begin to address this challenge, several studies have explored the use of supervised deep learning techniques using convolutional neural networks (CNNs) or transformers for improving cell segmentation accuracy (*Weigert et al., 2020*; *Stringer et al., 2021*; *Iqbal et al., 2019*; *Hörst et al., 2023*). Various methods now exist for performing post-hoc instance segmentation on the models' outputs in order to separate segmentation masks into individual cells.

Typically, these methods use a multi-step approach, first segmenting cells in 2D images, optionally performing instance segmentation, and then reconstructing them in 3D using the volume information (*Stringer et al., 2021*). While this can be successful in many contexts, this approach can suffer from low recall or have trouble retaining finer, non-convex labeling. Nevertheless, by training on (ideally large) human-annotated datasets, these supervised learning methods can learn to accurately segment cells in 2D, and ample 2D datasets now exist thanks to community efforts (*Ma et al., 2024*).

However, directly segmenting volumes in 3D ('direct-3D') could limit errors and streamline processing by retaining important morphological information (*Weigert et al., 2020*). Yet, 3D annotated datasets are lacking (*Ma et al., 2024*), likely due to the fact that they are highly time-consuming to generate. For example, to our knowledge, no 3D segmentation datasets of cells in whole-brain LSM volumes are available, despite the existence of open-source microscopy database repositories (*Williams et al., 2017*). Thus, here we provide the first human-annotated ground truth 3D data from mesoSPIM samples in over 2,5 K neural nuclei from the mouse neocortex. This data not only can be used for benchmarking algorithms as they emerge, but can be used in ongoing efforts to build foundation models for 3D microscopy.

While supervised deep learning is extremely powerful, it requires ample ground truth data which is often lacking. On the other hand, in computer vision, self-supervised learning (unsupervised learning) is emerging as a powerful approach for training deep neural networks without the need for explicit labeling of ground truth data. In the context of segmentation of cells, several studies have explored the use of unsupervised techniques to learn representations of cellular structures and improve segmentation accuracy (*Yao et al., 2022*; *Han and Yin, 2021*). However, these methods rely on adversarial learning, which can be difficult to train and have not been shown to provide accurate 3D results on cleared tissue for LSM data, which can suffer from clearing and other related artifacts.

Here, we developed a custom Python toolbox for direct-3D supervised and self-supervised cell segmentation built on state-of-the-art transformers and 3D CNN architectures (*Xia and Kulis, 2017*; *Hatamizadeh et al., 2022*) paired with classical image processing techniques (*Robert et al., 2020*). We benchmark our methods against Cellpose and StarDist - two leading supervised cell segmentation packages with user-friendly workflows - on our newly generated 3D ground truth dataset and show our supervised method can match or outperform them (in the low data regime) in 3D semantic segmentation on mesoSPIM-acquired volumes. Then, we show that our novel self-supervised model, which we named WNet3D, can be as good as or better than supervised models without any human-labeled data for training. Lastly, we benchmarked on three other diverse open-source 3D datasets, one acquired with LSM (Platynereis-Nuclei), and two others acquired with confocal imaging (Mouse-Skull-Nuclei and Platynereis-ISH-Nuclei) (*Lalit and Tomancak, 2021*).

## Results and Discussion

Whole mouse brain LSM followed by counting nuclei is becoming an increasingly popular task thanks to advances in imaging and tissue clearing techniques (*Chung et al., 2013*; *Renier et al., 2014*; *Ertürk, 2024*). Nuclear counting can be useful for c-FOS quantification, post-hoc verification of calcium indicator imaging location, and anatomical mapping. However, in order to develop more robust computer vision methods for these tasks, new 3D datasets must be developed, as none exist to date. Therefore, we developed a 3D human-annotated dataset based on data acquired with a mesoSPIM system (*Voigt et al., 2019*; *Figure 1a*, see Methods and the Dataset Card). Using whole-brain data from mice, we cropped small regions and human-annotated in 3D 2632 neurons that were endogenously labeled by TPH2-tdTomato (*Figure 1a*). In order to aid experts in performing labeling, we built a 3D annotator in napari, which is included in our Python package called CellSeg3D (see Methods).

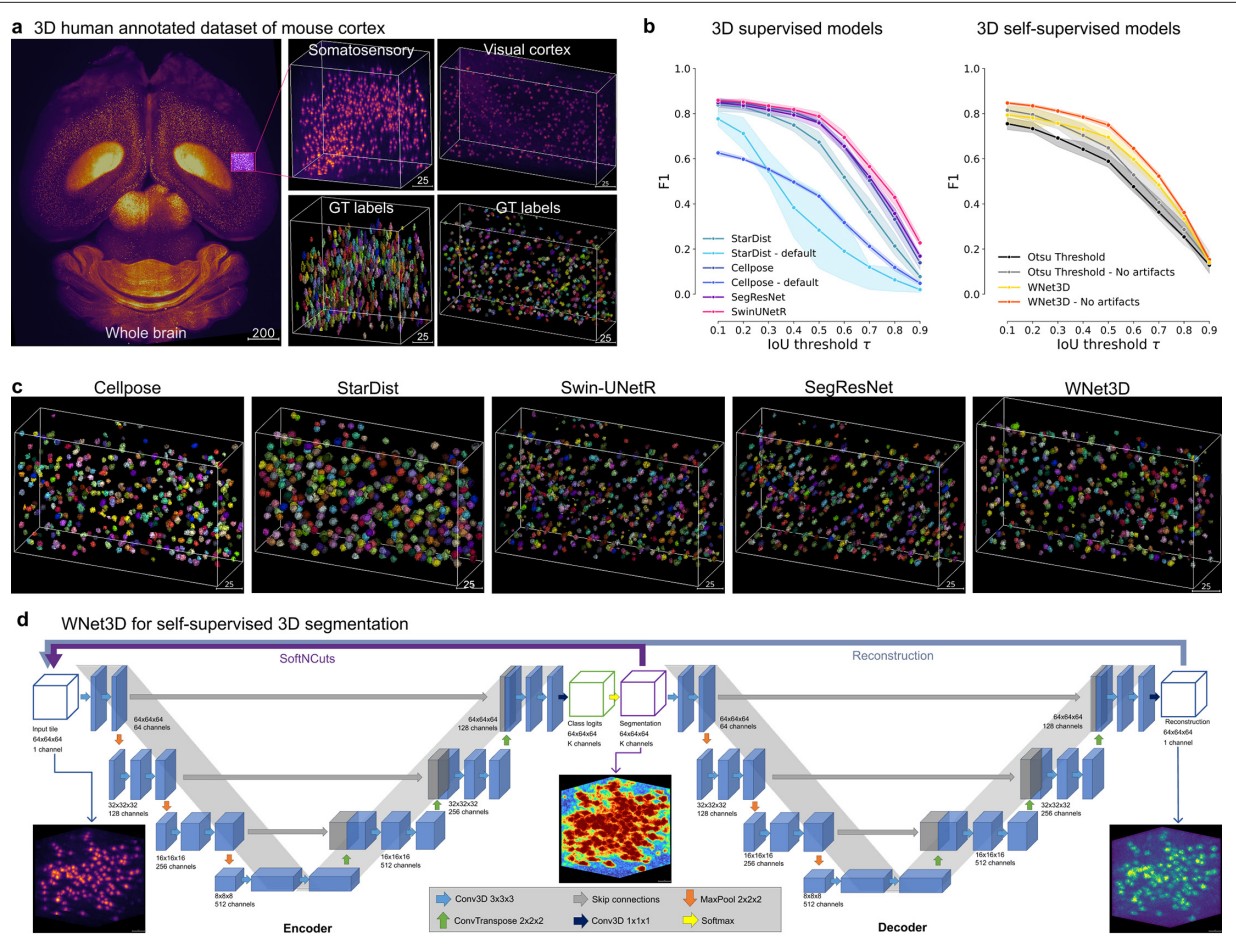

**Figure 1.** Performance of 3D semantic and instance segmentation models. (**a**) Raw mesoSPIM whole-brain sample, volumes, and corresponding ground truth labels from somatosensory (**S1**) and visual (**V1**) cortical regions. (**b**) Evaluation of instance segmentation performance for: baseline with Otsu thresholding only, supervised models: Cellpose, StartDist, SwinUNetR, SegResNet; and our self-supervised model WNet3D over three data subsets. F1-score is computed from the Intersection over Union (IoU) with ground truth labels, then averaged. Error bars represent 50% ~Confidence Intervals (CIs). (**c**) View of 3D instance labels from models, as noted, for the visual cortex volume. (**d**) Illustration of our WNet3D architecture showcasing the dual 3D U-Net structure with our modifications (see Methods).

The online version of this article includes the following figure supplement(s) for figure 1:

**Figure supplement 1.** Hyperparameter tuning of baselines and statistics.

**Figure supplement 2.** Training WNet3D: Overview of the training process of WNet3D.

To show performance on this new dataset, we benchmarked Cellpose (*Stringer et al., 2021*; *Pachitariu and Stringer, 2022*) and StarDist (*Weigert et al., 2020*). Cellpose is a spatial-embedding-based instance segmentation method. The network predicts a flow vector at each pixel, representing the pre-computed solution of the heat diffusion equation applied to instance masks, with the heat source at the object center. During inference, these learned flows guide pixel grouping, linking those that converge to the same location. Cellpose-3D extends Cellpose by using the trained 2D model, and processing all slices of a test volume independently along the xy, xz, and yz planes. This generates two estimates of the gradient in x, y, and z for each point (six total predictions), which are averaged to obtain a full set of 3D vector gradients. ROI generation then follows a 3D gradient vector tracking step, clustering pixels that converge to the same fixed points. StarDist predicts distances from each pixel (or voxel) to the boundary of the surrounding object along predefined directions (rays). This allows for precise instance segmentation, particularly for objects with star-convex shapes, making StarDist one of the most widely applied methods in this domain.

We then trained two additional models from different classes - transformers and 3D convolutional neural networks - for supervised direct-3D segmentation. Specifically, we leveraged a SwinUNetR

transformer (*Hatamizadeh et al., 2022*), and a SegResNet CNN (*Myronenko, 2018*) from the MONAI project (*The MONAI Consortium, 2020*). SwinUNetR is a transformer-based segmentation model that combines the Swin Transformer architecture with the UNet design. It leverages the self-attention mechanism of transformers for capturing long-range dependencies and multi-scale features in the input data. The hierarchical structure of the Swin transformer allows SwinUNetR to process images with variable resolutions efficiently. SegResNet is a convolutional neural network (CNN) developed for 3D medical image segmentation (*Myronenko, 2018*). It is based on a ResNet-like architecture, incorporating residual connections to improve gradient flow and model convergence during training. SwinUNetR and SegResNet are optimized for volumetric segmentation tasks but not used previously in cell segmentation tasks.

We found that our supervised models (SwinUNetR and SegResNet) have comparable instance segmentation performance to Cellpose and StarDist on held-out (unseen) test data set as measured by the F1 vs. IoU threshold (see Methods, *Figure 1b and c*) and thus are highly amendable to cell segmentation tasks. For a fair comparison, we performed a hyperparameter sweep of all the models we tested (*Figure 1—figure supplement 1a-d*), and in *Figure 1b and c* we show the quantitatively and qualitatively best models. We also compared to a non-deep learning-based baseline consisting of Otsu's method followed by Voronoi-Otsu instance segmentation to generate predictions (*Figure 1b*). Importantly, many deep learning-based models could achieve excellent performance on our new dataset (*Figure 1b and c*), with the SwinUNetR transformer performing the best (*Figure 1b*).

While supervised models are extremely powerful when labeled data is available to train on, in many new applications, there is limited to no human-annotated data. Thus, self-supervised methods can be highly attractive, as they require no human annotation. Self-supervised learning for 3D cell segmentation relies on the assumption that structural and morphological features of cells can be inferred directly from unlabeled data. This involves leveraging inherent properties such as spatial coherence and local contrast in imaging volumes to distinguish cellular structures. This approach assumes that meaningful representations of cellular boundaries and nuclei can emerge solely from raw 3D volumes. By modeling these properties, algorithms can be used across varied tissue conditions, including tissues that have some artifacts (i.e. from LSM and the clearing processes), but such artifacts may need a post-processing step to filter out extra large or small particles. To note, a strength of this approach is that self-supervised methods are better equipped to generalize across diverse imaging modalities and datasets by capturing underlying structural features, rather than relying on potentially biased human labels. Thus, as with any approach, it has its trade-offs.

Here, we built a new self-supervised model called WNet3D for direct-3D segmentation that requires no ground truth training data, only raw volumes. Our WNet3D model is inspired by WNet (*Xia and Kulis, 2017*) (see Methods, *Figure 1b and c*). Our changes include a conversion to a fully 3D architecture and adding the SoftNCuts loss, replacing the proposed two-step model update with the weighted sum of the encoder and decoder losses, and trimming the number of weights for faster inference (*Figure 1d*, *Figure 1—figure supplement 2a*, and see Methods). We found that WNet3D could be as good or better than the fully supervised models, especially in the low data regime, on this dataset at semantic segmentation (*Figure 2a*; averaged values across data splits are shown in *Figure 1—figure supplement 1e*, and statistical values are in *Figure 1—figure supplement 1f*).

Notably, our pre-trained WNet3D, which is trained on 100% of the raw data without any labels, achieves 0.81±0.004 F1-Score with simple filtering of artifacts (removing the slices containing the problematic regions; *Figure 1—figure supplement 1g*) and 0.74±0.12 without any filtering. To compare, we trained supervised models with 10, 20, 60, or 80% of the training data and tested on the held-out data subsets. Considering models with 80% of the training data, the F1-Score for SwinUNetR was 0.83±0.01, 0.76±0.03 for Cellpose (tuned), 0.74±0.006 for SegResNet, 0.72±0.007 for StarDist (tuned), 0.61±0.007 for StarDist (default), and 0.43±0.09 for Cellpose (default). For WNet3D with 80% raw data for training was 0.71±0.03 (unfiltered) (*Figure 2a*; an unfiltered example is shown in *Figure 1—figure supplement 1g*), which is still on-par with supervised models.

For models with only 10% of the training data, the F1-Score was 0.78±0.07 for SwinUNetR, 0.69±0.02 for StarDist (tuned), 0.42±0.13 for SegResNet, 0.39±0.36 for StarDist (default), 0.33±0.4 for Cellpose (tuned), 0.20±0.35 for Cellpose (default), and WNet3D was 0.74±0.02 (unfiltered), which is still on-par with the top supervised model, and much improved (2X) over most supervised baselines, most strikingly at low-data regimes (*Figure 2a*).

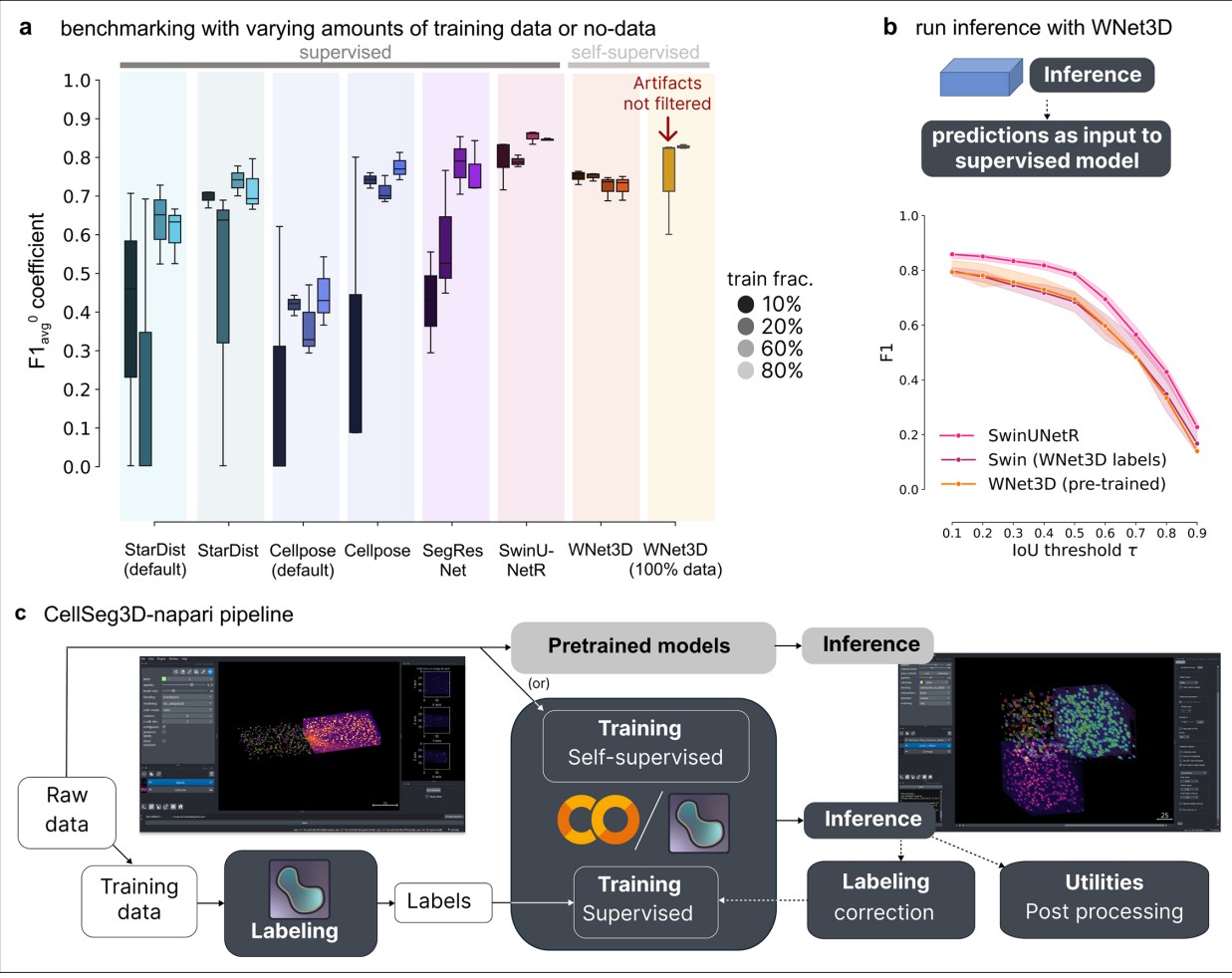

**Figure 2.** Benchmarking the performance of WNet3D vs.supervised models with various amounts of training data on our mesoSPIM dataset. (**a**) Semantic segmentation performance: comparison of model efficiency, indicating the volume of training data required to achieve a given performance level. Each supervised model was trained with an increasing percentage of training data (with 10, 20, 60, or 80%, left to right/dark to light within each model grouping, see legend); F1-Score score with an $IoU >= 0$ was computed on unseen test data, over three data subsets for each training/evaluation split. Our self-supervised model (WNet3D) is also trained on a subset of the training set of images, but always without ground truth human labels. Far right: We also show the performance of the pre-trained WNet3D available in the plugin (far right), with and without cropping the regions where artifacts are present in the image. See Methods for details. The central box represents the interquartile range (IQR) of values with the median as a horizontal line, the upper and lower limits the upper and lower quartiles. Whiskers extend to data points within 1.5 IQR of the quartiles. (**b**) Instance segmentation performance comparison of Swin-UNetR and WNet3D (pretrained, see Methods), evaluated on unseen data across 3 data subsets, compared with a Swin-UNetR model trained using labels from the WNet3D self-supervised model. Here, WNet3D was trained on separate data, producing semantic labels that were then used to train a supervised Swin-UNetR model, still on held-out data. This supervised model was evaluated as the other models, on 3 held-out images from our dataset, unseen during training. Error bars indicate 50% ~CIs. (**c**) Workflow diagram depicting the segmentation pipeline: either raw data can be used directly (self-supervised) or labeled and used for training, after which other data can be used for model inference. Each stream concludes with post-hoc inspection and refinement, if needed (post-processing analysis and/or refining the model).

Thus, on this new MesoSPIM 3D dataset (over the four different data subsets we tested), we find significant differences in model performance (Kruskal-Wallis H test, H=49.21, p=2.06e-08, n=12). With post-hoc Conover-Iman testing, WNet3D showed significant performance gains over StarDist and Cellpose (defaults) (statistics in *Figure 1—figure supplement 1f*). Importantly, it is not significantly different from the best-performing supervised models (i.e. SwinUNetR p=1, and other competitive supervised models: Cellpose (tuned) p=0.21, or SegResNet p=0.076; *Figure 1—figure supplement 1f*). Altogether, our self-supervised model can perform as well as top supervised approaches on this novel dataset.

As WNet3D is self-supervised, it therefore cannot inherently discriminate cells vs. artifacts – it has no notion of a 'cell.' Therefore, filtering can be used to clean up artifacts when sufficient (e.g. using

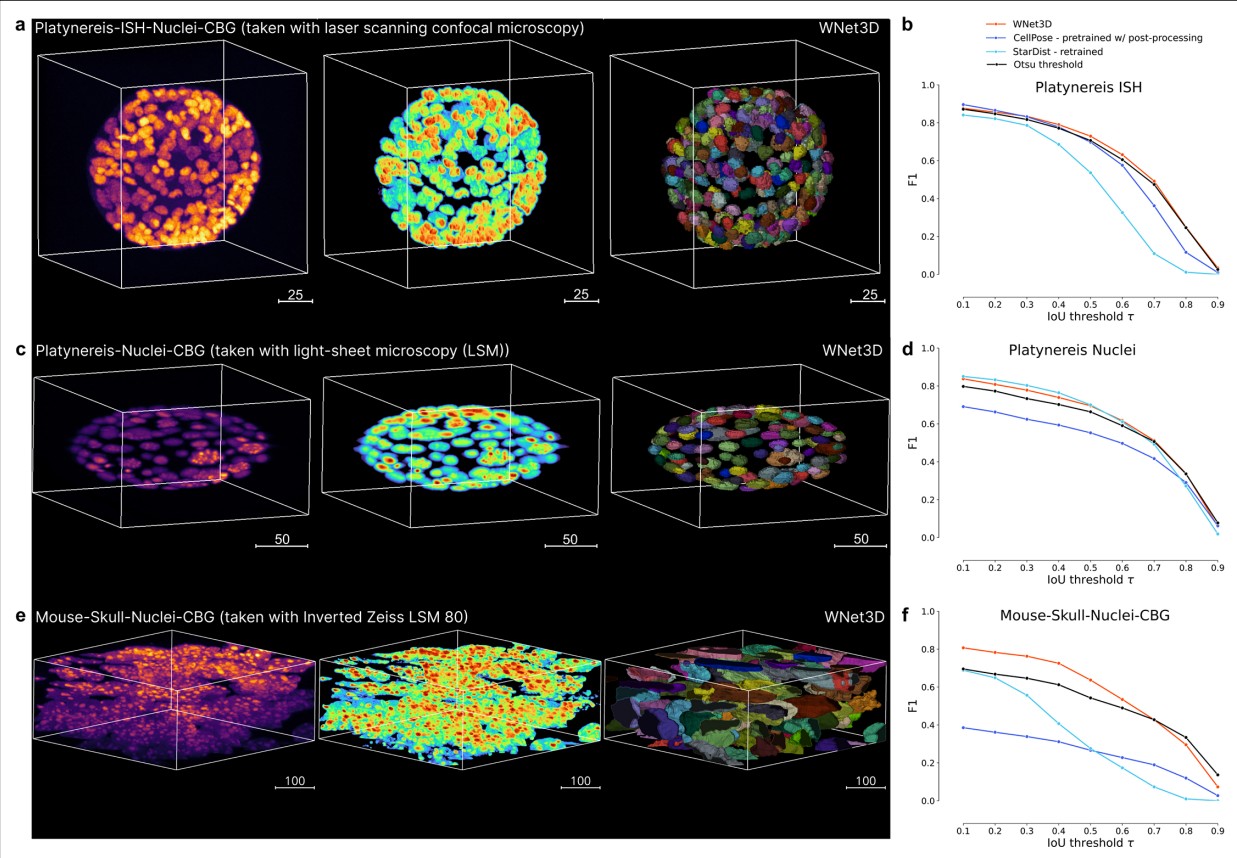

**Figure 3.** Benchmarking on additional datasets. (**a**) Left: 3D Platynereis-ISH-Nuclei confocal data; middle is WNet3D semantic segmentation; right is instance segmentation. (**b**) Instance segmentation performance (zero-shot) of the pretrained WNet3D, Otsu threshold, and supervised models (Cellpose, StarDist) on select datasets featured in **a**, shown as F1-score vs intersection over union (IoU) with ground truth labels. (**c**) Left: 3D Platynereis-Nuclei LSM data; middle is WNet3D semantic segmentation; right is instance segmentation. (**d**) Instance segmentation performance (zero-shot) of the pretrained WNet3D, Otsu threshold, and supervised models (Cellpose, StarDist) on select datasets featured in **c**, shown as F1-score vs IoU with ground truth labels. (**e**) Left: Mouse Skull-Nuclei Zeiss LSM 880 data; middle is WNet3D semantic segmentation; right is instance segmentation. A demo of using CellSeg3D to obtain these results is available here: https://www.youtube.com/watch?v=U2a9IbiO7nE&t=12s. (**f**) Instance segmentation performance (zero-shot) of the pretrained WNet3D, Otsu threshold, and supervised models (Cellpose, StarDist) on select datasets featured in **e**, shown as F1-score vs IoU with ground truth labels.

rules based on label volume to remove aberrantly small or large particles), or one could use WNet3D to generate 3D labels, correct them, and then use these semi-manually annotated images in order to train a suitable supervised model (such as Cellpose or SwinUNetR), which would be able to distinguish artifacts from cells. This process is called active learning, and can generally speed up data annotation (***Figure 2b***).

To show the feasibility of this approach, we trained a SwinUNetR using WNet3D self-supervised generated labels (***Figure 2b***) and show it can be nearly as good as a fully supervised model that required human 3D labels (no significant difference across F1 vs. IoU thresholds; Kruskal-Wallis H test H=4.91, p=0.085, n=9). Labeling, training, and this active inference learning can be completed in the CellSeg3D napari plugin we provide (***Figure 2c***). Moreover, the models we present are available in Jupyter Notebooks (which can be used locally or on Google Colab) and in a new napari plugin we developed, with full support for labeling, training (self-supervised or supervised), model inference, and evaluation plus other utilities (***Figure 2c***). We also provide our pretrained WNet3D model weights for user testing, and we further benchmark the model below.

We benchmarked WNet3D, Cellpose, StarDist, plus the non-deep learning baseline, on three other 3D datasets that were recently developed (***Lalit and Tomancak, 2021***). These three additional data-sets have varying cell sizes and cell density, and are collected with either LSM or confocal microscopy (***Figure 3a, c and e***). We used the pretrained Cellpose model (***Pachitariu and Stringer, 2022***), trained

**Table 1.** F1-Scores for additional benchmark datasets, where we test our pretrained WNet3D, zero-shot. Kruskal-Wallis H test [dataset, statistic, p-value]: Platynereis-ISH-Nuclei-CBG, 1.6, 0.69; Platynereis-Nuclei-CBG, 3.06, 0.38; Mouse-Skull-Nuclei-CBG (within post-processed), 10.13, **0.018**; Mouse-Skull-Nuclei-CBG (no processing), 15.8, **0.001**.

| | $F_{0.1}^1$ | $F_{0.2}^1$ | $F_{0.3}^1$ | $F_{0.4}^1$ | $F_{0.5}^1$ | $F_{0.6}^1$ | $F_{0.7}^1$ | $F_{0.8}^1$ | $F_{0.9}^1$ | $F_{MEAN}^1$ |
|---|---|---|---|---|---|---|---|---|---|---|
| **Platynereis-ISH-Nuclei-CBG:** | | | | | | | | | | |
| Otsu threshold | 0.872 | 0.847 | 0.817 | 0.772 | 0.706 | 0.605 | 0.474 | 0.246 | 0.026 | 0.596 |
| Cellpose (supervised) | <u>0.896</u> | <u>0.866</u> | 0.832 | 0.778 | 0.698 | 0.576 | 0.362 | 0.117 | 0.010 | 0.570 |
| StarDist (supervised) | 0.841 | 0.822 | 0.786 | 0.686 | 0.536 | 0.326 | 0.110 | 0.011 | 0. | 0.458 |
| WNet3D (zero-shot) | 0.876 | 0.856 | <u>0.834</u> | <u>0.790</u> | <u>0.729</u> | <u>0.632</u> | <u>0.492</u> | <u>0.249</u> | <u>0.034</u> | <u>0.610</u> |
| **Platynereis-Nuclei-CBG:** | | | | | | | | | | |
| Otsu threshold | 0.798 | 0.773 | 0.733 | 0.702 | 0.663 | 0.590 | 0.507 | 0.336 | 0.077 | 0.576 |
| Cellpose (supervised) | 0.691 | 0.663 | 0.624 | 0.594 | 0.553 | 0.497 | 0.417 | 0.290 | 0.062 | 0.488 |
| StarDist (supervised) | <u>0.850</u> | <u>0.833</u> | <u>0.803</u> | <u>0.764</u> | <u>0.700</u> | 0.611 | 0.492 | 0.272 | 0.019 | 0.594 |
| WNet3D (zero-shot) | 0.838 | 0.808 | 0.778 | 0.739 | 0.695 | <u>0.617</u> | <u>0.512</u> | <u>0.338</u> | <u>0.059</u> | <u>0.598</u> |
| **Mouse-Skull-Nuclei-CBG (most challenging dataset)** | | | | | | | | | | |
| Otsu threshold | 0.667 | 0.634 | 0.596 | 0.566 | 0.495 | 0.427 | 0.369 | 0.276 | 0.097 | 0.458 |
| Otsu threshold +post-processing | 0.695 | 0.668 | 0.647 | 0.612 | 0.543 | 0.490 | 0.428 | 0.334 | 0.137 | 0.506 |
| Cellpose (supervised) | 0.137 | 0.111 | 0.077 | 0.054 | 0.038 | 0.028 | 0.020 | 0.014 | 0.006 | 0.054 |
| Cellpose +post-processing | 0.386 | 0.362 | 0.339 | 0.312 | 0.266 | 0.228 | 0.189 | 0.120 | 0.027 | 0.248 |
| StarDist (supervised) | 0.573 | 0.533 | 0.411 | 0.253 | 0.135 | 0.065 | 0.020 | 0.003 | 0.0 | 0.221 |
| StarDist +post-processing | 0.689 | 0.649 | 0.557 | 0.407 | 0.276 | 0.174 | 0.073 | 0.010 | 0.0 | 0.315 |
| WNet3D (zero-shot) | 0.766 | 0.732 | 0.669 | 0.572 | 0.455 | 0.355 | 0.254 | 0.175 | 0.033 | 0.446 |
| WNet3D+post-processing | <u>0.807</u> | <u>0.783</u> | <u>0.763</u> | <u>0.725</u> | <u>0.637</u> | <u>0.534</u> | <u>0.428</u> | <u>0.296</u> | <u>0.073</u> | <u>0.561</u> |

a StarDist model (as no pretrained model existed), and used our pretrained WNet3D model that was only pretrained on the mesoSPIM dataset we presented above. Note, this is a strong test of generalization of our model, as it was only trained on a single dataset in a self-supervised manner. Our pre-trained WNet3D generalizes quite favorably on most datasets, and on average has the highest F1-Score on each individual dataset (*Table 1*, *Figure 3a–f*). Notably, WNet3D showed strong performance on the challenging Mouse Skull dataset (*Figure 3e and f*; *Figure 1—figure supplement 2b*).

Lastly, as a worked example, we tested our pre-trained WNet3D on mouse whole-brain tissue that was cleared and stained with cFOS then imaged with a mesoSPIM microscope (*Figure 4a and b*; see Methods). We used the BrainReg (*Tyson et al., 2022*; *Niedworok et al., 2016*; *Claudi et al., 2020*) registration toolkit to align our sample to the Allen Institute Brain Atlas (https://mouse.brain-map.org/). We then selected brain regions (such as motor cortex) using our CellSeg3D package, and ran model inference (*Figure 4b*).

## Discussion and limitations

One major limitation for the field has been the lack of 3D data (*Ma et al., 2024*). We provide the first-in-kind open source ground truth dataset of mesoSPIM mouse brain data that we hope sparks more

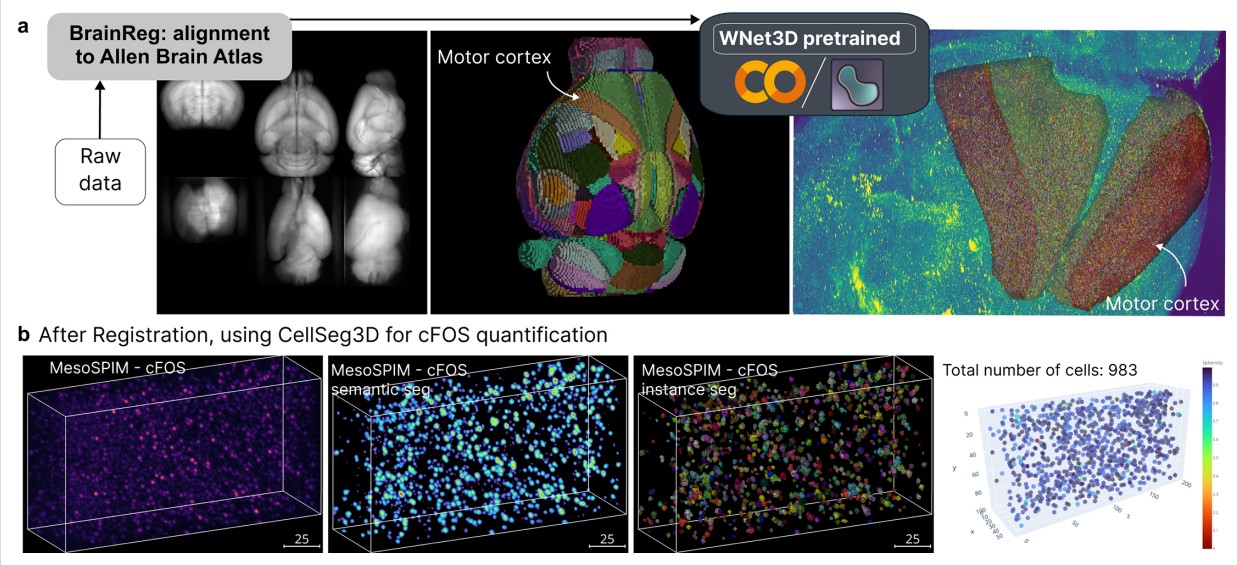

**Figure 4.** CellSeg3D napari plugin example outputs. (**a**) Demo using a cleared and MesoSPIM imaged c-FOS mouse brain, followed by BrainReg (20.22) registration to the Allen Brain Atlas https://mouse.brain-map.org/, then processing of regions of interest (ROIs) with CellSeg3D. Here, the WNet3D was used for semantic segmentation followed by processing for instance segmentation. (**b**) Qualitative example of WNet3D-generated prediction (thresholded) and labels on a crop from the c-FOS-labeled whole-brain. A demo of using CellSeg3D to obtain these results is available here: https://www.youtube.com/watch?v=3UOvvpKxEAo.

methods to be developed. Thus, while we put considerable efforts here to provide a new neuron 3D dataset, more datasets will be needed in the future to understand the limitations of self-supervised learning for this type of data and beyond.

Another limitation is that self-supervised methods are going to excel in samples that have enough separation in the signal-to-noise (i.e. clearly visible nuclei). As discussed above, our method works by detection, and as with any semantic segmentation method, this then requires fine-tuning of threshold parameters. With ground truth data, this is straightforward, but if one lacks any ground truth, this can be subjective. Yet, setting the threshold often can be largely guided from the scientific question at hand. Therefore, while tuning such a parameter is required (which is equally the case for, i.e. Cellpose pre-trained models), with the tooling we provide, the threshold becomes easier to set based on visual inspection of the objects of interest, as long as the objects in the volumes respect the previously mentioned assumptions. We aimed to limit this problem by showing how active learning can help by using this approach to generate reasonable labels for downstream fine-tuning. Namely, in *Figure 2b*, we show how self-supervised learning can act as a step towards pseudo-labeling. We provide the software to then inspect and correct these pseudo-labels. These labels can then be used for training, achieving performance on par-with top supervised methods, such as the SwinUNetR transformer.

While we focused our efforts on rather uncluttered nuclei -- except for the challenging mouse skull in *Figure 3e* where WNet3D performs better than supervised models -- we believe that our self-supervised semantic segmentation model could generalize to other fluorescence 3D data as it becomes available, despite the limitations. However, we have never tested our approach on electron microscopy data, or for axon tracing, so other tools are likely to be more suitable for those tasks (*Dorkenwald et al., 2023*; *Friedmann et al., 2020*). For instance segmentation, if the cells are more overlapping, etc., more complex methods, such as the star-convex polygons used by StarDist to approximate the shapes of cell nuclei, could be adapted to recover higher-quality instance labels (since it is agnostic to the backbone used *Weigert et al., 2020*). Nonetheless, we believe that the benefit of fully self-supervised learning is worth the cost of post-hoc processing for these types of easy-to-spot and fix mistakes, given that generating a large ground truth 3D dataset is on the order of hundreds of human-hours of labeling efforts.

**Table 2.** Dataset ground-truth cell count per volume.

| Region | Size | Count |
|---|---|---|
| | (pixels) | (# of cells) |
| Sensorimotor | | |
| 1 | 199 × 106 × 147 | 343 |
| 2 | 299 × 78 × 111 | 365 |
| 3 | 299 × 105 × 147 | 631 |
| 4 | 249 × 93 × 114 | 396 |
| 5 | 249 × 86 × 94 | 347 |
| Visual | 329 × 127 × 214 | 485 |

## Conclusions

In summary, the CellSeg3D Python package supports high-performance supervised and self-supervised direct-3D segmentation for quantifying cells, as shown on four benchmark datasets. Our napari plugin supports both our new pretrained WNet3D for testing, the ability to train the WNet3D, and to use other top supervised models presented here (SegResNet, SwinUNetR). We also provide various tools for pre- and post-processing as well as utilities for labeling in 3D. We additionally provide our new 2632 cell 3D dataset intended for benchmarking 3D cell segmentation algorithms on meso-SPIM acquired cleared tissue (see Dataset Card). All code and data is fully open-sourced at https://github.com/AdaptiveMotorControlLab/CellSeg3D.

## Methods

### Datasets

#### CellSeg3D mesoSPIM dataset: Acquisition and labeling

The whole-brain data by *Voigt et al., 2019* was obtained from the IDR platform (*Williams et al., 2017*); the volume consists of CLARITY-cleared tissue from a TPH2-tdTomato mouse. Data was acquired with the mesoSPIM system at a zoom of 0.63 X with 561 nm excitation.

The data was cropped to several regions of the somatosensory (five volumes, without artifacts) and visual cortex (one volume, with artifacts) and annotated by an expert. All volumes were annotated by hand (see **Dataset Card** below for more details). The ground-truth cell count for the dataset is as follows (*Table 2*):

#### Additional benchmarking datasets from EmbedSeg

Additional datasets, used in *Figure 3* were used from the GitHub page of EmbedSeg (*Lalit and Tomancak, 2021*; *JugLab, 2021*). We used our pretrained WNet3D, without re-training (the model was only trained on our new MesoSPIM dataset described above), to generate semantic segmentation. Images and labels were first cropped to contents, discarding empty regions on the edges. We then downscaled the images and labels by a factor of two to reduce runtime. We obtain the raw WNet3D prediction simply by adding the images to napari, and using the Inference tool of the plugin with WNet3D, without changing any parameters from default. Note that usually one would enable thresholding, window inference, and instance segmentation in the napari GUI to directly obtain usable instance segmentation, however, this is also possible in Jupyter Notebooks, which we used for reproducibility to create the results shown.

Next, the channel containing the foreground was thresholded and the Voronoi-Otsu method from pyclEsperanto (*Robert et al., 2020*) was used to generate instance labels (for Platynereis data), with hyperparameters based on the F1-Score metric with the ground truth from data separate to the one on which we evaluate performance. However, these parameters can also be estimated directly. This is documented here.

For the Mouse Skull Nuclei instance segmentation, we performed additional post-processing using pyclEsperanto (*Robert et al., 2020*) to perform a morphological closing operation with radius 8 on

semantic labels in order to remove small holes. The image was then remapped to values $\in [0; 100]$ for convenience, before merging labels with a touching border within intensity range between 35 and 100 using the *merge_labels_with_border_intensity_within_range* function. This is documented in our linked Figures here.

We additionally report for these datasets the performance of the latest pretrained 'nuclei' Cellpose model, and a retrained a StarDist model (as no suitable pretrained model existed). For Cellpose, the object size parameter was estimated using the provided size model in the GUI, and the 'nuclei' pre-trained model was run to obtain instance labels. Other parameters were kept to defaults. For StarDist, models were trained with all remaining data in the dataset (i.e. excluding volumes used to report performance), as a training set with an 80%/20% train/validation split. All parameters and data augmentation used were the defaults, aside from training patch size, which was set to (64, 64, 64), which let all objects fit within the field of view. NMS and object thresholds were optimized after training using the provided functions. For inference on Mouse-Skull-Nuclei-CBG, the tiled prediction mode was used to allow volumes to fit in memory. We show performance on Mouse Skull with and without the extra post-processing, as well as a qualitative example of the effect of the post-processing (*Figure 3*, *Figure 1—figure supplement 2b*).

### c-FOS dataset
For the MesoSPIM c-FOS demo, we used a wild-type C57BL/6 J adult mouse (17 wk old, Female) that was given appetitive food 90 min before deep anesthesia and intra-cardial perfusion with 4% PFA. We followed established guidelines for iDISCO (*Renier et al., 2014*). In brief, the brain was dehydrated, bleached, permeabilized, and stained for c-FOS using anti-c-FOS Rat monoclonal purified IgG (Synaptic Systems, Cat. No. 226 017) followed by a Donkey anti-Rat IgG Alexa Fluor 555 (Invitrogen A78945) secondary antibody. Then, the whole brain was imaged on a mesoSPIM (*Voigt et al., 2019*). Imaging was performed with a laser at a wavelength of 561 nm, with a pixel size of 5.26 × 5.26 µm in x,y, and a step size of 5 µm in z. All experimental protocols adhered to the stringent ethical standards set forth by the Veterinary Department of the Canton Geneva, Switzerland, with all procedures receiving approval and conducted under license number 33020 (GE10A).

## Segmentation models and algorithms: Self-supervised semantic segmentation
### WNet3D model architecture
To perform self-supervised cell segmentation, we adapted the WNet architecture proposed by *Xia and Kulis, 2017*, an autoencoder architecture based on joining two U-Net models end-to-end. We provide a modified version of the WNet, named WNet3D, with the following changes:

- A conversion of the architecture for fully 3D segmentation, including a 3D SoftNCuts loss
- Replacing the proposed two-step model update with the weighted sum of the encoder and decoder losses, updated in a single backward pass
- Reducing the overall depth of the encoder and decoder, using three up/downsampling steps instead of four
- Replacing batch normalization with group normalization, tuning the number of groups based on performance

Reducing the number of layers improved overall performance by reducing overfitting and sped up training and inference. This trimming was meant to reduce the large number of parameters resulting from a naive conversion of the original WNet architecture to 3D, which were found to be unnecessary for the present cell segmentation task. Finally, we introduced group normalization (*Wu and He, 2018*) to replace batch normalization, which improved performance in the present low batch size setting, as well as training and inference speed.

To summarize, the model consists of an encoder $U_{enc}$ and decoder $U_{dec}$, as originally proposed; however, each UNet comprises seven blocks, for a total of 14 blocks, down from nine blocks per UNet originally. $U_{enc}$ and $U_{dec}$ start and end with 2 3×3×3 3D convolutional layers, in between are five blocks, each block being defined by two 3×3×3 3D convolutional layers, followed by a ReLU and group normalization (*Wu and He, 2018*) (instead of batch normalization). Skip connections are used to propagate information by concatenating the output of descending blocks to that of their corresponding

ascending blocks. Each block is followed by 2×2×2 max pooling layers in the descending half of $U_{enc}$ and $U_{dec}$, the ascending half uses 2×2×2 transpose convolution layers with stride=2; $U_{enc}$ is then followed by a 1×1×1 3D convolutional layer to obtain class logits, followed by a softmax, the output of which is provided to $U_{dec}$ to perform the reconstruction. $U_{dec}$ is similarly followed by a 1×1×1 3D convolutional layer and outputs the reconstructed volume. Refer to *Figure 1* for an overview of the WNet3D architecture.

## Losses

Segmentation is performed in $U_{enc}$ by using an adapted 3D SoftNCuts loss (*Shi and Malik, 2000*) as an objective, with the voxel brightness differences defining the edge weight in the calculation, as proposed in the initial Ncuts algorithm.

The SoftNCuts is defined as

$$Ncut_K(V) = \sum_{k=1}^{K} \frac{cut(A_k, V - A_k)}{cut(A_k, V)} \tag{1}$$

where $cut(A, B) = \sum_{u \in A, v \in B} w(u, v)$, $V$ is the set of all pixels, $A_k$ the set of all pixels labeled as class $k$ ($K$ being the number of classes, which is set to 2 here) and $w(u, v)$ is the weight of the edge $uv$ in a graph representation of the image. In order to group the voxels according to brightness, $w(u, v)$ is defined here as

$$w(u, v) = e^{\frac{-\|F(u) - F(v)\|_2^2}{\sigma_I}} * \begin{cases} e^{\frac{-\|X(u) - X(v)\|_2^2}{\sigma_X}} & \text{if } \|X(u) - X(v)\| < r \\ 0 & \text{otherwise} \end{cases} \tag{2}$$

with $F(i) = I(i)$ the intensity value, $X$ the spatial position of the voxel, $\sigma_I$ the standard deviation of the feature similarity term, termed 'intensity sigma,' $\sigma_X$ the standard deviation of the spatial proximity term, termed 'spatial sigma,' and $r$ the radius for the calculation of the loss, to avoid computing every pairwise value.

In our experiments, lowering the radius greatly sped up training without impacting performance, even with a radius as low as 2 voxels. For the spatial sigma, the original value of 4 was used, whereas for the intensity sigma, we use a value of 1 (originally 4), after remapping voxel values in each image to the [0; 100] range.

$U_{dec}$ then uses a suitable reconstruction loss to reconstruct the original image; we used either Mean Squared Error (MSE) or Binary Cross Entropy (BCE) as defined in PyTorch.

## WNet3D hyperparameters

To achieve proper cell segmentation, it was crucial to prevent the SoftNCuts loss from simply separating the data in broad regions with differing overall brightness; this was achieved by adjusting the weighting of the reconstruction loss accordingly. In our experiments, we empirically adapted the weights to equalize the contribution of each loss term, ensuring balanced gradients in the backward pass. This proved effective for training on our provided dataset; however, for different samples, adjusting the reconstruction weight and learning rate using the ranges specified below was necessary for good performance; other parameters were kept constant.

The default number of classes is two, to segment background and cells, but this number may be raised to add more brightness-grouped classes; this could be useful to mitigate the over-segmentation of cells due to brightness 'halos' surrounding the nucleus, or to help produce labels for object boundary segmentation.

We found that summing the losses, instead of iteratively updating the encoder first followed by the whole network as suggested, improved stability and consistency of loss convergence during training; in our version, the trade-off between accuracy of reconstruction and quality of segmentation is controlled by adjusting the parameters of the weighted sum instead of individual learning rates.

This modified model was usually trained for 50 epochs, unless stated otherwise. We use a batch size of 2, 2 classes, a radius of 2 for the NCuts loss and the MSE reconstruction loss, and use a

learning rate between $2\times10^{-3}$ and $2\times10^{-5}$ and reconstruction loss weight between $5\times10^{-3}$ and $5\times10^{-1}$, depending on the data.

See *Figure 1—figure supplement 2a* for an overview of the training process, including loss curves and model outputs.

## Segmentation models and algorithms: Supervised semantic segmentation

### Model architectures

In order to perform supervised fully 3D cell segmentation, we leveraged computer vision models and losses implemented by the MONAI project, which offers several state-of-the-art architectures. The MONAI API was used as the basis for our napari plugin, and we retained two of the provided models based on their performance on our dataset:

- SegResNet (*Myronenko, 2018*. 3D MRI brain tumor segmentation using autoencoder regularization, November 2018. *arXiv*. http://arxiv.org/abs/1810.11654)
- SwinUNetR (*Hatamizadeh et al., 2022*. UNETR: Transformers for 3D Medical Image Segmentation. [WACV]. http://arxiv.org/abs/2103.10504)

SegResNet is based on the CNN architecture, whereas SwinUNetR uses a transformer-based encoder.

Several relevant segmentation losses are made available for training:

- Dice loss (*Milletari et al., 2016*)
- Dice-Cross Entropy loss
- Generalized Dice loss (*Sudre et al., 2017*)
- Tversky loss (*Salehi et al., 2017*)

The SegResNet and SwinUNetR models shown here were trained using the Generalized Dice loss for 50 epochs, with a learning rate of $1\times10^{-3}$, batch size of 5 (SwinUNetR) or 10 (SegResNet), and data augmentation enabled. Unless stated otherwise, a train/test split of 80/20% was used.

The outputs were then passed through a threshold to discard low-confidence predictions; this was estimated using the training set to find the threshold that maximized the Dice metric between predictions and ground truth. Using the training set for this process ensures that we do not overfit the evaluation set on which we calculate the metrics. See the following notebook for the corresponding code: here. The 'Find best threshold' utility in the napari plugin allows one to perform this search immediately between a pair of labels and prediction volumes. We provide a full demo of how to estimate thresholds on a case-by-case basis in the following video: https://www.youtube.com/watch?v=xYbYqL1KDYE. The same process was repeated for Cellpose (for cell probability threshold) and StarDist (non-maximum suppression (NMS) and cell probability thresholds) to ensure fair comparisons, see 'Model comparison' below and *Figure 1*, *Figure 1—figure supplement 1*a, b, c, d for tuning results. Inference outputs are processed a posteriori to obtain instance labels, as detailed below.

### Instance segmentation

Several methods for instance segmentation are available in the plugin: the connected components and watershed algorithms (scikit-image), and the Voronoi-Otsu labeling method (clEsperanto). The latter combines an Otsu threshold and a Voronoi tessellation to perform instance segmentation, and more readily avoids fusing clumped cells than the former two, provided that the objects are convex, which is the case in the present task.

The Voronoi-Otsu method was, therefore, used to perform instance segmentation in the benchmarks, with its two parameters, spatial sigma and outline sigma, tuned to fit the training data when relevant, and manually selected otherwise.

## Model comparisons

StarDist was retrained using the provided example notebook for 3D, using default parameters. For the model we refer to as 'Default,' we used a patch size of $8\times 64 \times 64$, a grid of (2,1,1) , a batch size of 2 and 96 rays, as computed automatically in the provided example code for StarDist. For the 'Tuned'

version (referred to simply as 'StarDist'), we changed the patch size to 64 × 64 × 64 and the grid to (1,1,1).

Cellpose was retrained without pretrained weights using default parameters, except for the mean diameter which was set to 3.3 according to the provided object size estimation utility in the GUI (and visually confirmed). We investigated pretrained models provided by Cellpose, as well as attempting transfer learning, but no pretrained model was found to be suitable for our data. Despite Cellpose automatically resizing the data to match its training data, neither the automated estimate of object size, nor fixing the object size value manually helped in improving performance, therefore, we retrained those models with our data. 'Default' refers to automatically estimated parameters for StarDist (NMS and probability threshold, estimated on the training data), and cell probability threshold of 0 with resampling enabled for Cellpose. For both models, inference hyperparameters (respectively NMS and cell probability threshold for StarDist and cell probability threshold and resampling on CellPose) were tuned on the training set to maximize the F1-Score/Dice metric with GT labels, exactly like our models. After tuning, we found that Cellpose achieved best performance with a cell probability threshold of −9 and resampling enabled (see *Figure 1—figure supplement 1a* and here) across all data subsets. For StarDist, best parameters varied across subsets (see *Figure 1—figure supplement 1d* and here), however, as this did not affect performance significantly, we used the parameters estimated automatically as part of the training.

Models provided in the plugin (SwinUNetR, SegResNet, and WNet3D), which we refer to as 'pretrained,' are trained on the entire MesoSPIM dataset to obtain best possible performance, using all images (and labels only for the supervised models). The WNet3D model was used in *Figure 1b* (WNet3D), *Figure 2* (WNet3D pre-trained), and *Figure 3b, d, f* (WNet3D). Hyperparameters used are as mentioned above, except for the number of epochs, which was selected based on validation performance to avoid overfitting.

For *Figure 1b*, we trained each model on a subset of the dataset (sensorimotor volumes), chunked into 64 pixels cubes using an 80/20% training/validation split, and estimated the best threshold on the same training data. Next, we used the remaining held-out data (visual volume) to evaluate performance. Code for thresholds optimization may be found here, and code to create *Figure 2* can be found here. We also compared the performance of all models with that of a non-learning based thresholding, by using Otsu's threshold method followed by the Voronoi-Otsu instance segmentation function from pyClEsperanto to generate predictions. When comparing these results obtained by Otsu threshold with WNet3D results in *Figure 1b*, we additionally report performance on a specific subset of volumes without regions containing artifacts, without any differences in post-processing across methods.

## Label efficiency comparison

To assess how many labeled cells are required to reach a certain performance, we trained StarDist, Cellpose, SegResNet, SwinUNetR, and WNet3D on three distinct subsets of the data, each time holding out one full volume of the full dataset for evaluation, fragmenting the remaining volumes and labels into 64-pixel cubes, and training on distinct train/validation splits on remaining data. We used 10, 20, 60, and 80% splits in order to assess how much labeled data is necessary for the supervised models, and whether they show variability based on the data used for training. Of note, the evaluation data remained the same for all percentages in a given data subset, ensuring a consistent performance comparison. We used 50 epochs for all runs, and no early stopping or hyperparameter tuning was performed based on the validation performance during training. Instead, we reused the best hyperparameters found for *Figure 1b*.

For example, the first subset consists of all five somatosensory cortex volumes as training/validation data, and the visual cortex volume is held out for evaluation. For Cellpose, two conditions are shown, default (cell probability threshold of 0) and fine-tuned (threshold of −9), which improved performance.

To avoid training on data with artifacts present in the visual cortex volume, WNet3D was only trained on the first of the subsets. Instead, the model was trained on a percentage of the first subset using three different seeds. We also avoid evaluating on artifacts in the visual volume, as the model is not meant to handle these regions. It should be noted that this filtering does not consist of any additional post-processing on the volume, but strictly on cropping out regions with artifacts before evaluation.

Instance labels were generated as stated above, and then converted back to semantic labels to compute the F1-Score, see Performance evaluation section below.

## WNet3D-based retraining of supervised models

To assess whether WNet3D can generalize to unseen data when trained on a specific brain volume, we trained a WNet3D from scratch using volumes cropped from a different mesoSPIM-acquired whole-brain sample, labeled with c-FOS, imaged at 561 nm with a pixel size of $5.26 \times 5.26 \, \mu m$ in x and y, and a step size in z of $5 \, \mu m$ (see Additional datasets).

This model was then used to generate labels for our provided dataset. A SwinUNetR model was then trained using these WNet3D generated labels, and compared to the performance of the pretrained model we provide in our napari plugin.

## Performance evaluation

The models were evaluated using standard segmentation metrics (*Hirling et al., 2024*), namely F1-Score and intersection over union (*IoU*). The equations for these evaluation metrics are shown below, with TP, FP, and FN representing true positives (TP), false positives (FP), and false negatives (FN), respectively. The higher the F1 (precision and recall), the better the model performance.

$$\text{IoU} = \frac{\text{TP}}{\text{TP+FP+FN}}, \quad \text{F1-Score} = \frac{2\text{TP}}{2\text{TP+FP+FN}}, \quad \text{Precision} = \frac{\text{TP}}{\text{TP+FP}}, \quad \text{Recall} = \frac{\text{TP}}{\text{TP+FN}}$$

We used the evaluation utilities provided by StarDist (*Weigert et al., 2020*).

To assess performance for semantic segmentation, we report the F1-Score without any *IoU* threshold, which is then equivalent to the Dice score computed on the semantic labels, given the Boolean nature of the data.

The metric to assess instance segmentation accuracy can be computed as functions of several overlap thresholds; true positives are pairings of model predictions and ground-truth labels having an intersection over union (*IoU*) value greater than the specified threshold, with automated matching to prevent additional instances from being assigned to the same ground truth or model-predicted instance of a label. We report the F1-Score over a range of *IoU* thresholds between 0.1 and 0.9 (step size of 0.1).

For instance segmentation, we take the model's probability outputs and apply an intensity threshold to get semantic predictions; this threshold ultimately affects the reported metrics, therefore, we discuss the procedure here. We set these thresholds based on the training set. Specifically, to determine the optimal threshold for evaluating instance segmentation on a training fold, pairs of predictions and corresponding labels from the training set were taken. For each pair, the threshold that maximized the F1-Score at $IoU >= 0$, which is equivalent to the Dice coefficient, was computed. This process was repeated for all images within the training fold. The resulting optimal thresholds provided the threshold used when evaluating that particular fold. The code for each use case can be found here. For the mesoSPIM data, this threshold was empirically found to be 0.4 (SwinUNetR), 0.3 (SegResNet), and 0.6 (WNet3D), in *Figure 2*. For *Figure 3*, the thresholds for WNet3D were: 0.45 for Mouse Skull and 0.55 for both the Platynereis datasets. We then convert these thresholded results to instance labels using the Voronoi-Otsu algorithm, the parameters of which were chosen based on the F1-Score metric between ground truth labels and model-generated labels on the training set, as described in the Model Section above describing instance segmentation. If a model is not trained, i.e., for example in *Figure 3b*, we set these parameters manually to threshold by eye. To reproduce the F1-Scores as shown, we used the following values (*Table 3*):

**Table 3.** Parameters used for instance segmentation with the pyclEsperanto Voronoi-Otsu function.

| Dataset | Outline σ | Spot σ |
| --- | --- | --- |
| mesoSPIM | 0.65 | 0.65 |
| Mouse Skull | 1 | 15 |
| Platynereis-ISH | 0.5 | 2 |
| Platynereis | 0.5 | 2.75 |

## CellSeg3D napari plugin workflow

To facilitate the use of our models, we provide a napari plugin where users can easily annotate data, train models, run inference, and perform various post-processing steps. Starting from raw data, users can quickly crop regions into regions of interest, and create training data from those. Users may manually annotate the data in napari using our labeling interface, which provides additional interfaces such as orthogonal projections to better view the ongoing labeling process, as well as keeping track of time spent labeling each slice, or alternatively train a self-supervised model to automatically perform a first iteration of the segmentation and labeling, without annotation. Users can also try pretrained models, including the self-supervised one, to generate labels which can then be corrected using the same labeling interface. Supervised or self-supervised models can then be trained using the generated data. Full documentation for the plugin can be found on our Github website.

In the case of supervised learning, the volumes (random patches or whole images) are split into training and validation sets according to a user-set proportion, using 80%/20% by default. Input images are normalized by setting all values above and below the 1st and 99th percentiles to the corresponding percentile value, respectively. Data augmentation can be used; by default a random shift of the intensity, elastic and affine deformations, flipping, and rotation are used.

For the self-supervised model, images are remapped to values in the [0;100] range to accommodate the intensity sigma of the SoftNCuts loss. No percentile normalization is used and data augmentation is restricted to flipping and rotating in this case.

Deterministic training may also be enabled for all models and the random generation seed set; unless specified otherwise, models were trained on cropped cubes with 64 pixels edges, with both data augmentation and deterministic training enabled.

We additionally provide a Colab Notebook to train our self-supervised model using the same procedure described above. The pretrained weights for all our models are also made available through the HuggingFace platform (and automatically downloaded by the plugin or in Colab), so that users without the recommended hardware can still easily train or try our models. All code is open source and available on https://github.com/AdaptiveMotorControlLab/CellSeg3D (copy archived at *Achard et al., 2025*).

## Statistical methods

To confirm whether there were statistically significant differences in model performance, we pooled accuracy values across IoU, and/or across percentage of training data used. We used Python 3.8–3.10 using the https://pypi.org/project/scikit-posthocs/ package, and we performed a Kruskal-Wallis test to check the null hypothesis that the median of all models was equal. When this test was significant, we used two-sided Conover-Iman post-hoc testing to test pairwise differences between models, also using the 'scikit_posthoc' implementation, with the Holm-Bonferroni correction for multiple comparisons (step-down method using Bonferroni adjustments).

## Acknowledgements

The authors thank Martin Weigert, Jessy Lauer, and members of the Mathis Lab for inputs and comments on the manuscript. MWM acknowledges the Vallee Foundation and the Wyss Institute for partly funding this work. MWM is the Bertarelli Foundation Chair for Integrative Neuroscience.

## Additional information

### Funding

| Funder | Grant reference number | Author |
| --- | --- | --- |
| Vallee Foundation | MATHIS | Timokleia Kousi<br>Maxime Vidal<br>Mackenzie Weygandt Mathis |

| Funder | Grant reference number | Author |
|---|---|---|
| Wyss Center for Bio and Neuroengineering | MATHIS | Maxime Vidal<br>Asim Iqbal<br>Stéphane Pagès<br>Mackenzie Weygandt Mathis |

The funders had no role in study design, data collection and interpretation, or the decision to submit the work for publication.

## Author contributions

Cyril Achard, Conceptualization, Software, Formal analysis, Investigation, Visualization, Methodology, Writing – original draft; Timokleia Kousi, Data curation, Validation, Investigation; Markus Frey, Supervision, Methodology; Maxime Vidal, Conceptualization, Software, Methodology; Yves Paychere, Colin Hofmann, Methodology; Asim Iqbal, Data curation, Methodology; Sebastien B Hausmann, Data curation, Formal analysis, Investigation, Visualization; Stéphane Pagès, Resources, Data curation, Supervision, Investigation; Mackenzie Weygandt Mathis, Conceptualization, Resources, Software, Supervision, Funding acquisition, Visualization, Writing – original draft, Project administration, Writing – review and editing, Methodology

## Author ORCIDs

Cyril Achard (ID) https://orcid.org/0009-0003-6992-6928
Markus Frey (ID) https://orcid.org/0000-0003-0291-3391
Asim Iqbal (ID) https://orcid.org/0000-0003-2174-4554
Stéphane Pagès (ID) https://orcid.org/0000-0003-0618-777X
Mackenzie Weygandt Mathis (ID) https://orcid.org/0000-0001-7368-4456

## Ethics

All experimental protocols adhered to the stringent ethical standards set forth by the Veterinary Department of the Canton Geneva, Switzerland, with all procedures receiving approval and conducted under license number 33020 (GE10A).

Reviewer #1 (Public review): https://doi.org/10.7554/eLife.99848.4.sa1
Reviewer #2 (Public review): https://doi.org/10.7554/eLife.99848.4.sa2
Author response https://doi.org/10.7554/eLife.99848.4.sa3

---

# Additional files

## Supplementary files

MDAR checklist

## Data availability

Labeled 3D data are available at: https://doi.org/10.5281/zenodo.11095110; see our Supplemental Data Card. All code is available at: https://github.com/AdaptiveMotorControlLab/CellSeg3D, (copy archived at *Achard et al., 2025*) and code to reproduce the Figures is available at: https://github.com/C-Achard/CellSeg3D-figures (copy archived at *Achard and Mathis, 2025*).

The following dataset was generated:

| Author(s) | Year | Dataset title | Dataset URL | Database and Identifier |
|---|---|---|---|---|
| Mathis Laboratory of Adaptive Intelligence | 2024 | 3D ground truth annotations of cleared whole mouse brain nuclei imaged with a mesoSPIM system | https://doi.org/10.5281/zenodo.11095111 | Zenodo, 10.5281/zenodo.11095111 |

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

## Appendix 1

### Dataset card

A. Motivation

1. *For what purpose was the dataset created? Was there a specific task in mind? Was there a specific gap that needed to be filled? Please provide a description.*

The contributions of our dataset to the vision and cell biology communities are twofold: (1) We release a 3D cell segmentation dataset of 2632 TPH2 positive cells, based on data from *Voigt et al., 2019*. (2) It is entirely human-annotated. The dataset is one of the first cell segmentation datasets to date created in 3D.

2. *Who created the dataset (which team, research group) and on behalf of which entity (company, institution, organization)?*

The human-annotated dataset was created by the Mathis Lab of Adaptive Intelligence of EPFL, who are co-authors of this work. The raw brain data is publicly available on https://idr.openmicroscopy.org/webclient/?show=project-854.

3. *Who funded the creation of the dataset? If there is an associated grant, please provide the name of the grantor and the grant name and number.*

This project was funded, in part, by the Wyss Center via a grant to PI Mathis.

4. *Any other comments?* No.

### Composition

1. *What do the instances that comprise the dataset represent (e.g. documents, photos, people, countries)? Are there multiple types of instances (e.g. movies, users, and ratings; people and interactions between them; nodes and edges)? Please provide a description.*

The instances in our dataset represent 3D volumetric segments, extracted from mesoSPIM scans of mouse brains. Each instance is essentially a three-dimensional image that has been carefully hand-cropped mainly from the somatosensory and visual cortex of the scanned data. In each of these 3D volumes, TPH2 cells are identified and labeled.

2. *How many instances are there in total (of each type, if appropriate)?*

There are six 3D volumetric segments that contain a total of 2638 TPH2 positive cells identified and labeled in 3D.

3. *Does the dataset contain all possible instances or is it a sample (not necessarily random) of instances from a larger set? If the dataset is a sample, then what is the larger set? Is the sample representative of the larger set (e.g. geographic coverage)? If so, please describe how this representativeness was validated/verified. If it is not representative of the larger set, please describe why not (e.g. to cover a more diverse range of instances, because instances were withheld or unavailable).*

The dataset provided is a subset of the available whole-brain sample, selected from larger raw volumetric data obtained from mesoSPIM scans of mouse brains. This selection primarily consists of 3D volumes cropped mainly from the somatosensory and visual cortex regions, where the TPH2 cells are labeled meticulously. The broader dataset from which these instances were extracted represents scans of whole mouse brains. However, due to the immense volume of the entire scanned data, creating a manageable and focused dataset was key for addressing specific research questions and computational manageability.

4. *What data does each instance consist of? 'Raw' data (e.g. unprocessed text or images) or features? In either case, please provide a description.*

Each instance in the dataset consists of 'raw' 3D volumetric data derived from mesoSPIM scans of mouse brains, specifically focusing on the somatosensory cortex and vision cortex regions. The instances are essentially unprocessed and maintain the integrity of the original scanned data.

5. *Is there a label or target associated with each instance? If so, please provide a description.*

Yes, each instance in the dataset is human-annotated with masks. There are no categories or text associated with the masks.

6. *Is any information missing from individual instances? If so, please provide a description, explaining why this information is missing (e.g. because it was unavailable). This does not include intentionally removed information, but might include, e.g., redacted text.*

In our dataset, there is no information missing from individual instances.

7. *Are relationships between individual instances made explicit (e.g. users' movie ratings, social network links)? If so, please describe how these relationships are made explicit.*

Not applicable.

8. *Are there any errors, sources of noise, or redundancies in the dataset? If so, please provide a description.*

While we have taken extensive measures to ensure the accuracy and quality of the dataset, it is challenging to rule out the presence of minor errors or noise, especially considering the complex nature of the 3D cell segmentation task. Nonetheless, we believe that any such inconsistencies do not compromise the overall reliability and utility of the dataset.

9. *Is the dataset self-contained, or does it link to or otherwise rely on external resources (e.g. websites, tweets, other datasets)? If it links to or relies on external resources, (a) are there guarantees that they will exist, and remain constant, over time; (b) are there official archival versions of the complete dataset (i.e. including the external resources as they existed at the time the dataset was created); (c) are there any restrictions (e.g. licenses, fees) associated with any of the external resources that might apply to a dataset consumer? Please provide descriptions of all external resources and any restrictions associated with them, as well as links or other access points, as appropriate.*

The dataset is self-contained.

10. *Does the dataset contain data that might be considered confidential (e.g. data that is protected by legal privilege or by doctor-patient confidentiality, data that includes the content of individuals' non-public communications)? If so, please provide a description.*

No.

11. *Does the dataset contain data that, if viewed directly, might be offensive, insulting, threatening, or might otherwise cause anxiety? If so, please describe why.*

No. The dataset is composed solely on scientific, non-human biological data.

12. *Does the dataset identify any subpopulations (e.g. by age, gender)? If so, please describe how these subpopulations are identified and provide a description of their respective distributions within the dataset.*

Not applicable.

13. *Is it possible to identify individuals (i.e. one or more natural persons), either directly or indirectly (i.e. in combination with other data) from the dataset? If so, please describe how.*

Not applicable.

14. *Does the dataset contain data that might be considered sensitive in any way (e.g. data that reveals race or ethnic origins, sexual orientations, religious beliefs, political opinions or union memberships, or locations; financial or health data; biometric or genetic data; forms of government identification, such as social security numbers; criminal history)? If so, please provide a description.*

No.

15. *Any other comments?*

No.

## Collection process

1. *How was the data associated with each instance acquired? Was the data directly observable (e.g. raw text, movie ratings), reported by subjects (e.g. survey responses), or indirectly inferred/derived from other data (e.g. part-of-speech tags, model-based guesses for age or language)? If the data was reported by subjects or indirectly inferred/derived from other data, was the data validated/ verified? If so, please describe how.*

The data associated with each instance was acquired through mesoSPIM scans of mouse brains, providing raw, directly observable 3D volumetric data. The data was not reported by subjects or indirectly inferred or derived from other data; it was directly observed and recorded from the scientific imaging process. All collected volumes were annotated by expert human annotators. The quality of the annotations was validated by an external expert not involved in the annotation process.

2. *What mechanisms or procedures were used to collect the data (e.g. hardware apparatuses or sensors, manual human curation, software programs, software APIs)? How were these mechanisms or procedures validated?*

The raw data is open source and provided by the Image Data Resource (IDR).

*3. If the dataset is a sample from a larger set, what was the sampling strategy (e.g. deterministic, probabilistic with specific sampling probabilities)?*

Our sampling strategy was designed to select volumes where TPH2 cells are clearly discernible. We aimed to include a varied range of volumes, from densely packed with TPH2 cells to ones more sparsely populated, ensuring a good representation of various brain areas. Another important factor was the manageability of the volumes from an annotation perspective, to facilitate accurate and efficient labeling.

*4. Who was involved in the data collection process (e.g. students, crowdworkers, contractors) and how were they compensated (e.g. how much were crowdworkers paid)?*

The released masks were created by research personnel of the Mathis Lab of Adaptive Intelligence, EPFL.

*5. Over what timeframe was the data collected? Does this timeframe match the creation timeframe of the data associated with the instances (e.g. recent crawl of old news articles)? If not, please describe the timeframe in which the data associated with the instances was created.*

The raw data was downloaded from the Image Data Resource (IDR) website. The labels were created between June and October 2021.

If the dataset does not relate to people, you may skip the remaining questions in this section.

## Preprocessing/Cleaning/Labeling

*1. Was any preprocessing/cleaning/labeling of the data done (e.g. discretization or bucketing, tokenization, part-of-speech tagging, SIFT feature extraction, removal of instances, processing of missing values)? If so, please provide a description. If not, you may skip the remaining questions in this section.*

Yes, extensive preprocessing, and labeling were conducted to ensure the usability and reliability of the dataset. The initial step involved examination of the raw 3D volumetric data, where we ruled out the presence of anomalies or artifacts. During this phase, we ensured the visibility of TPH2-positive cells within the volumetric segments. We proceeded to label the TPH2-positive cells through a well-defined annotation process, where each cell within the selected volumes was identified and marked by our experts. At the end of the annotation process, the quality of the work was verified by a human expert not involved in the annotation work.

*2. Was the 'raw' data saved in addition to the preprocessed/cleaned/labeled data (e.g. to support unanticipated future uses)? If so, please provide a link or other access point to the 'raw' data.*

The raw data is open source and available on the Image Data Resource (IDR) website.

*3. Is the software that was used to preprocess/clean/label the data available? If so, please provide a link or other access point.*

Yes. We used the napari interactive viewer for multidimensional images in Python and used our plugin.

## Uses

*1. Has the dataset been used for any tasks already? If so, please provide a description.*

The dataset was used to train segmentation models.

*2. Is there a repository that links to any or all papers or systems that use the dataset? If so, please provide a link or other access point.*

Yes, the repository hosting the model weights which were trained on our data, as well as the repository for our napari plugin for 3D cell segmentation.

*3. What (other) tasks could the dataset be used for?*

We intend the dataset to be used to train 3D cell segmentation models. However, we invite the research community to gather additional annotations for mesoSPIM-acquired datasets via the tools we contribute in the present publication.

*4. Is there anything about the composition of the dataset or the way it was collected and preprocessed/cleaned/labeled that might impact future uses? For example, is there anything that a dataset consumer might need to know to avoid uses that could result in unfair treatment of individuals or groups (e.g. stereotyping, quality of service issues) or other risks or harms (e.g. legal risks, financial harms)? If so, please provide a description. Is there anything a dataset consumer could do to mitigate these risks or harms?*

Not applicable.

*5. Are there tasks for which the dataset should not be used? If so, please provide a description.*

Full terms of use for the dataset can be found at https://github.com/AdaptiveMotorControlLab/CellSeg3D, copy archived at *Achard et al., 2025*. The project is made open source under an MIT license.

## Distribution

*1. Will the dataset be distributed to third parties outside of the entity (e.g. company, institution, organization) on behalf of which the dataset was created? If so, please provide a description.*

The dataset is released on Zenodo at: https://zenodo.org/records/11095111.

*2. How will the dataset be distributed (e.g. tarball on website, API, GitHub)? Does the dataset have a digital object identifier (DOI)?*

The dataset is released on Zenodo at: https://zenodo.org/records/11095111.

*3. When will the dataset be distributed?*

The dataset is released on Zenodo at: https://zenodo.org/records/11095111 alongside the publication of this paper.

*4. Will the dataset be distributed under a copyright or other intellectual property (IP) license, and/ or under applicable terms of use (ToU)? If so, please describe this license and/or ToU, and provide a link or other access point to, or otherwise reproduce, any relevant licensing terms or ToU, as well as any fees associated with these restrictions.*

The dataset is released under an MIT license.

*5. Have any third parties imposed IP-based or other restrictions on the data associated with the instances? If so, please describe these restrictions, and provide a link or other access point to, or otherwise reproduce, any relevant licensing terms, as well as any fees associated with these restrictions.*

Full terms of use and restrictions on use of the provided 3D cell segmentation dataset can be found at https://github.com/AdaptiveMotorControlLab/CellSeg3D , copy archived at *Achard et al., 2025*.

*6. Do any export controls or other regulatory restrictions apply to the dataset or to individual instances? If so, please describe these restrictions, and provide a link or other access point to, or otherwise reproduce, any supporting documentation.*

The dataset is released under an MIT license.

*7. Any other comments?*

No.

## Maintenance

*1. Who will be supporting/hosting/maintaining the dataset?*

The dataset is available at https://zenodo.org/records/11095111 and maintained by the Mathis Lab of Adaptive Intelligence.

*2. How can the owner/curator/manager of the dataset be contacted (e.g. email address)?*

Please see contact information at https://github.com/AdaptiveMotorControlLab/CellSeg3D, copy archived at *Achard et al., 2025* or write to Mackenzie Mathis: mackenzie.mathis@epfl.ch.

*3. Is there an erratum? If so, please provide a link or other access point.*

No.

*4. Will the dataset be updated (e.g. to correct labeling errors, add new instances, delete instances)? If so, please describe how often, by whom, and how updates will be communicated to dataset consumers (e.g. mailing list, GitHub)?*

To ensure reproducibility of research, this dataset won't be updated. Any issues or errors will be publicly shared.

*5. If the dataset relates to people, are there applicable limits on the retention of the data associated with the instances (e.g. were the individuals in question told that their data would be retained for a fixed period of time and then deleted)? If so, please describe these limits and explain how they will be enforced.*

Not applicable.

*6. Will older versions of the dataset continue to be supported/hosted/maintained? If so, please describe how. If not, please describe how its obsolescence will be communicated to dataset consumers.*

This is the first version.

*7. If others want to extend/augment/build on/contribute to the dataset, is there a mechanism for them to do so? If so, please provide a description. Will these contributions be validated/verified? If so, please describe how. If not, why not? Is there a process for communicating/distributing these contributions to dataset consumers? If so, please provide a description.*

We warmly encourage users to enhance the value of this project by contributing additional annotations and annotated datasets. If you have relevant data, please consider sharing it by linking the data to our GitHub repository. For any inquiries, suggestions, or discussions related to the project, please feel free to reach out to us on GitHub https://github.com/AdaptiveMotorControlLab/CellSeg3D, copy archived at *Achard et al., 2025*.

*8. Any other comments?*

No.

## Data annotation card

### Task formulation

*1. At a high level, what are the subjective aspects of your task?*

Object segmentation within an image is a subjective task (*Kirillov et al., 2023*). Distinguishing between structures that represent cells and artifacts relies on the annotator's judgment and expertise. This can lead to variability in the quality and quantity of the masks generated per image by different annotators. To mitigate this risk, we engaged experts from our research lab to annotate the volumes. We insisted on the quality of annotations over their quantity; we aimed to annotate smaller volumes to ensure accurate representation of the cell nuclei, even if it meant having fewer annotations.

*2. What assumptions do you make about annotators?*

Our annotator is a member of our research lab, ensuring a close understanding of the project's goals. The team concentrated on two main objectives. (1) Clear Understanding of Project Goals: We worked to fully understand the project's aims and translated them into clear and straightforward guidelines, which included visual examples. (2) Regular Sharing of Updates and Results: We reviewed our aims and results to make ongoing improvements to the annotation process. This regular check-in helped in quickly addressing any issues and adding new material to improve our annotation quality.

*3. How did you choose the specific wording of your task instructions? What steps, if any, were taken to verify the clarity of task instructions and wording for annotators?*

The annotator was a co-creator of the annotation instructions and guidelines, which boosted their understanding. As our task was annotations images, we crafted visual examples with step-by-step instructions. We collectively decide how to handle complex and unambiguous cases, and refine the guidelines throughout the process. The project team met for feedback and updates, while the annotator was able to give feedback in an asynchronous way at any time.

*4. What, if any, risks did your task pose for annotators and were they informed of the risks prior to engagement with the task?*

No identified risks.

*5. What are the precise instructions that were provided to annotators?*

We created clear guides on installing and using the napari annotation tool. The task was to segment every TPH2-positive cell in a given image. The annotator created a 3D mask for each cell they identified, using the tool to precisely add or remove areas of the mask around the cell. In simpler terms, they had to isolate each cell in 3D using the tool, making sure it was accurate down to the pixel level.

### Selecting annotations

*1. Are there certain perspectives that should be privileged? If so, how did you seek these perspectives out?*

We chose to engage researchers that have a deep understanding of cell biology and vision research.

*2. Are there certain perspectives that would be harmful to include? If so, how did you screen these perspectives out?*

No.

*3. Were sociodemographic characteristics used to select annotators for your task? If so, please detail the process.*

No.

*4. If you have any aggregated socio-demographic statistics about your annotator pool, please describe. Do you have reason to believe that sociodemographic characteristics of annotators may have impacted how they annotated the data? Why or why not?*

Our annotator worked in our research institute.

*5. Consider the intended context of use of the dataset and the individuals and communities that may be impacted by a model trained on this dataset. Are these communities represented in your annotator pool?*

Not applicable.

## Platform and infrastructure choices

1. *What annotation platform did you utilize? At a high level, what considerations informed your decision to choose this platform? Did the chosen platform sufficiently meet the requirements you outlined for annotator pools? Are any aspects not covered?*

We used napari, a fast, interactive viewer for multi-dimensional images in Python. Link: https://napari.org/stable/

2. *What, if any, communication channels did your chosen platform offer to facilitate communication with annotators? How did this channel of communication influence the annotation process and/or resulting annotations?*

Communication was established through other internal communication platforms.

3. *How much were annotators compensated? Did you consider any particular pay standards, when determining their compensation? If so, please describe.*

The compensation was based on their employment contract at EPFL.

## Dataset analysis and evaluation

1. *How do you define the quality of annotations in your context, and how did you assess the quality in the dataset you constructed?*

To assess the quality of the annotations in the constructed dataset, we included a review process. Annotations were created by an expert well-acquainted with the morphological characteristics of TPH2-positive cells, ensuring a high level of initial accuracy. Any ambiguous cases in annotation were resolved through discussions amongst the team until a consensus was reached. Regular feedback was provided to the annotator, and any identified errors or inconsistencies were promptly corrected.

2. *Have you conducted any analysis on disagreement patterns? If so, what analyses did you use and what were the major findings? Did you analyze potential sources of disagreement?*

We provided regular feedback sessions in a synchronous and asynchronous way.

3. *How do the individual annotator responses relate to the final labels released in the dataset?*

Our dataset, along with our annotations are available and accessible through Zenodo: https://zenodo.org/records/11095111.

## Dataset release and maintenance

1. *Do you have reason to believe the annotations in this dataset may change over time? Do you plan to update your dataset?* No.

2. *Are there any conditions or definitions that, if changed, could impact the utility of your dataset?* We do not believe so.

3. *Will you attempt to track, impose limitations on, or otherwise influence how your dataset is used? If so, how?* No.

4. *Were annotators informed about how the data is externalized? If changes to the dataset are made, will they be informed*? Yes.

5. *Is there a process by which annotators can later choose to withdraw their data from the dataset? If so, please detail.* No.

