## [Editor Report · eLife Assessment]

This **important** work presents a self-supervised method for the segmentation of 3D cells in fluorescent microscopy images, conveniently packaged as a Napari plugin and tested on an annotated dataset. The segmentation method is **solid** and compares favorably to other learning-based methods and Otsu thresholding on four datasets, offering the possibility of eliminating time-consuming data labeling to speed up quantitative analysis. This work will be of interest to a wide variety of laboratories analysing fluorescently labeled images.

---

## [Referee Report · Reviewer #1 (Public review)]

The manuscript now compares the WNet3D quantitatively against other methods on all four datasets:

Figure 1b shows results on the mouse cortex dataset, comparing StarDist, CellPose, SegResNet, SwinUNetR against self-supervised (or learning-free methods) WNet3D and Otsu thresholding.

Figure 2b shows results on an unnamed dataset (presumably the mouse cortex dataset), comparing StarDist, CellPose, SegResNet, SwinUNetR with different levels of training data against WNet3D.

Figure 3 shows results on three datasets (Platynereis-ISH-Nuclei-CBG, Platynereis-Nuclei-CBG, and Mouse-Skull-Nuclei-CBG), comparing StarDist, CellPose against WNet3D and Otsu thresholding.

It is unclear whether the Otsu thresholding baseline was given the same post-processing as the WNet3D. Figure 1b shows two versions for WNet3D ("WNet3D - No artifacts" and "WNet3D"), but only one for Otsu thresholding. Given that post-processing (or artifact removal) seems to have a substantial impact on accuracy, the authors should clarify whether the Otsu thresholding results were treated in the same way and if Otsu thresholding was not post-processed. Figure 2a would also benefit from including the thresholding results (with and without artifact removal).

---

## [Referee Report · Reviewer #2 (Public review)]

The authors have now addressed the most important points, and they include more comprehensive evaluation of their method and comparisons to other approaches for multiple datasets.

Some points would benefit from clarification:

- Figure 1B now compares "Otsu thresholding", "WNet 3D - No artifacts" and "WNet 3d". Why don't you also report the score for "Otsu thresholding - No Artifacts"? To my understanding this is a post-processing operation to remove small and very large objects, so it could easily be applied to the Otsu thresholding. Given the good results for Otsu thresholding alone (quite close F1-score to WNet 3d), it seems like DL might not really be necessary at all for this dataset and including "Otsu thresholding - No artifacts" would enable evaluating this point.

- CellPose and StarDist perform poorly in all the experiments performed by the authors. In almost all cases they underperform Otsu thresholding, which is in most cases on par with the WNet results (except for "Mouse Skull Nuclei CBG"). This is surprising and contradicts the collective expertise of the community: good CellPose and StarDist models can be trained for the 3D instance segmentation tasks studied here. Perhaps these methods were not trained in an optimal way. Seems unlikely that it is not possible to get much better CellPose or StarDist models for these tasks (current versions are on par or much worse than Otsu!), as I have applied both of these models successfully in similar settings. Specifically, it seems unlikely that the developers of CellPose or StarDist would obtain similarly poor scores on the same data (note I am not one of the developers).

The current experiments still highlight an interesting aspect: the problem of training / fine-tuning these methods correctly on new data and the technical challenges associated with this. But the reported results should by no means be taken as a fair assessment of the capabilities of StarDist or CellPose.

Please note that I did not have time to test the Napari plugin again, so I did not evaluate whether it improved in usability.

---

## [Author Response]

The following is the authors’ response to the previous reviews

**eLife Assessment**
This work presents a valuable self-supervised method for the segmentation of 3D cells in microscopy images, alongside an implementation as a Napari plugin and an annotated dataset. While the Napari plugin is readily applicable and promises to eliminate time consuming data labeling to speed up quantitative analysis, there is incomplete evidence to support the claim that the segmentation method generalizes to other light-sheet microscopy image datasets beyond the two specific ones used here.

Technical Note: We showed the utility of CellSeg3D in the first submission and in our revision on 5 distinct datasets; 4 of which we showed F1-Score performance on. We do not know which “two datasets” are referenced. We also already showed this is not limited to LSM, but was used on confocal images; we already limited our scope and changed the title in the last rebuttal, but just so it’s clear, we also benchmark on two non-LSM datasets.

In this revision, we have now additionally extended our benchmarking of Cellpose and StarDrist on all 4 benchmark datasets, where our Wet3D (our novel contribution of a self-supervised model) outperforms or matches these supervised baselines. Moreover, we perform rigorous testing of our model’s generalization by training on one dataset and testing generalization to the other 3; we believe this is on par (or beyond) what most cell segmentation papers do, thus we hope that “incomplete” can now be updated.

**Public Reviews:**

**Reviewer #1 (Public review):**
This work presents a self-supervised method for the segmentation of 3D cells in microscopy images, an annotated dataset, as well as a napari plugin. While the napari plugin is potentially useful, there is insufficient evidence in the manuscript to support the claim that the proposed method is able to segment cells in other light-sheet microscopy image datasets than the two specific ones used here.

Thank you again for your time. We benchmarked already on four datasets the performance of WNet3Dd (our 3D SSL contribution) - thus, we do not know which two you refer to. Moreover, we now additionally benchmarked Cellpose and StarDist on all four so readers can see that on all datasets, WNet3D outperforms or matches these supervised methods.

I acknowledge that the revision is now more upfront about the scope of this work. However, my main point still stands: even with the slight modifications to the title, this paper suggests to present a general method for self-supervised 3D cell segmentation in light-sheet microscopy data. This claim is simply not backed up.

We respectfully disagree; we benchmark on four 3D datasets: three curated by others and used in learning ML conference proceedings, and one that we provide that is a new ground truth 3D dataset - the first of its kind - on mesoSPIM-acquired brain data. We believe benchmarking on four datasets is on par (or beyond) with current best practices in the field. For example, Cellpose curated one dataset and tested on held-out test data on this one dataset (https://www.nature.com/articles/s41592-020-01018-x) and benchmarked against StarDist and Mask R-CNN (two models). StarDist (Star-convex Polyhedra for 3D Object Detection and Segmentation in Microscopy) benchmarked on two datasets and against two models, IFT-Watershed and 3D U-Net. Thus, we feel our benchmarking on more models and more datasets is sufficient to claim our model and associated code is of interest to readers and supports our claims (for comparison, Cellpose’s title is “Cellpose: a generalist algorithm for cellular segmentation”, which is much broader than our claim).

I still think the authors should spell out the assumptions that underlie their method early on (cells need to be well separated and clearly distinguishable from background). A subordinate clause like "often in cleared neural tissue" does not serve this purpose. First, it implies that the method is also suitable for non-cleared tissue (which would have to be shown). Second, this statement does not convey the crucial assumptions of well separated cells and clear foreground/background differences that the method is presumably relying on.

We expanded the manuscript now quite significantly. To be clear, we did show our method works on non-cleared tissue; the Mouse Skull, 3D platynereis-Nuclei, and 3D platynereis-ISH-Nuclei is not cleared tissue, and not all with LSM, but rather with confocal microscopy. We attempted to make that more clear in the main text.

Additionally, we do not believe it needs to be well separated and have a perfectly clean background. While we removed statements like "often in cleared neural tissue", expanded the benchmarking, and added a new demo figure for the readers to judge. As in the last rebuttal, we provide video-evidence (https://www.youtube.com/watch?v=U2a9IbiO7nE) of the WNet3D working on the densely packed and hard to segment by a human, Mouse Skull dataset and linked this directly in the figure caption.

We have re-written the main manuscript in an attempt to clarify the limitations, including a dedicated “limitations” section. Thank you for the suggestion.

It does appear that the proposed method works very well on the two investigated datasets, compared to other pre-trained or fine-tuned models. However, it still remains unclear whether this is because of the proposed method or the properties of those specific datasets (namely: well isolated cells that are easily distinguished from the background). I disagree with the authors that a comparison to non-learning methods "is unnecessary and beyond the scope of this work". In my opinion, this is exactly what is needed to proof that CellSeg3D's performance can not be matched with simple image processing.

We want to again stress we benchmarked WNet3D on four datasets, not two. But now additionally added benchmarking with Cellpose, StarDist and a non-deep learning method as requested (see new Figures 1 and 3).

As I mentioned in the original review, it appears that thresholding followed by connected component analysis already produces competitive segmentations. I am confused about the authors' reply stating that "[this] is not the case, as all the other leading methods we fairly benchmark cannot solve the task without deep learning". The methods against which CellSeg3D is compared are CellPose and StarDist, both are deep-learning based methods.That those methods do not perform well on this dataset does not imply that a simpler method (like thresholding) would not lead to competitive results. Again, I strongly suggest the authors include a simple, non-learning based baseline method in their analysis, e.g.: * comparison to thresholding (with the same post-processing as the proposed method) * comparison to a normalized cut segmentation (with the same post-processing as the proposed method)

We added a non-deep learning based approach, namely, comparing directly to thresholding with the same post hoc approach we use to go from semantic to instance segmentation. WNet3D (and other deep learning approaches) perform favorably (see Figure 2 and 3).

Regarding my feedback about the napari plugin, I apologize if I was not clear. The plugin "works" as far as I tested it (i.e., it can be installed and used without errors). However, I was not able to recreate a segmentation on the provided dataset using the plugin alone (see my comments in the original review). I used the current master as available at the time of the original review and default settings in the plugin.

We updated the plugin and code for the revision at your request to make this possible directly in the napari GUI in addition to our scripts and Jupyter Notebooks (please see main and/or `pip install --upgrade napari-cellseg3d`’ the current is version 0.2.1). Of course this means the original submission code (May 2024) will not have this in the GUI so it would require you to update to test this. Alternatively, you can see the demo video we now provide for ease: https://www.youtube.com/watch?v=U2a9IbiO7nE (we understand testing code takes a lot of time and commitment).

We greatly thank the review for their time, and we hope our clarifications, new benchmarking, and re-write of the paper now makes them able to change their assessment from incomplete to a more favorable and reflective eLife adjective.

**Reviewer #2 (Public review):**
Summary:The authors propose a new method for self-supervised learning of 3d semantic segmentation for fluorescence microscopy. It is based on a WNet architecture (Encoder / Decoder using a UNet for each of these components) that reconstructs the image data after binarization in the bottleneck with a soft n-cuts clustering. They annotate a new dataset for nucleus segmentation in mesoSPIM imaging and train their model on this dataset. They create a napari plugin that provides access to this model and provides additional functionality for training of own models (both supervised and self-supervised), data labeling and instance segmentation via post-processing of the semantic model predictions. This plugin also provides access to models trained on the contributed dataset in a supervised fashion.Strengths:- The idea behind the self-supervised learning loss is interesting.- It provides a new annotated dataset for an important segmentation problem.- The paper addresses an important challenge. Data annotation is very time-consuming for 3d microscopy data, so a self-supervised method that yields similar results to supervised segmentation would provide massive benefits.- The comparison to other methods on the provided dataset is extensive and experiments are reproducible via public notebooks.Weaknesses:The experiments presented by the authors support the core claims made in the paper. However, they do not convincingly prove that the method is applicable to segmentation problems with more complex morphologies or more crowded cells/nuclei.Major weaknesses:(1) The method only provides functionality for semantic segmentation outputs and instance segmentation is obtained by morphological post-processing. This approach is well known to be of limited use for segmentation of crowded objects with complex morphology. This is the main reason for prediction of additional channels such as in StarDist or CellPose. The experiments do not convincingly show that this limitation can be overcome as model comparisons are only done on a single dataset with well separated nuclei with simple morphology. Note that the method and dataset are still a valuable contribution with this limitation, which is somewhat addressed in the conclusion. However, I find that the presentation is still too favorable in terms of the presentation of practical applications of the method, see next points for details.

Thank you for noting the methods strengths and core features. Regarding weaknesses, we have revised the manuscript again and added direct benchmarking now on four datasets and a fifth “worked example” (https://www.youtube.com/watch?v=3UOvvpKxEAo&t=4s) in a new Figure 4.

We also re-wrote the paper to more thoroughly present the work (previously we adhered to the “Brief Communication” eLife format), and added an explicit note in the results about model assumptions.

(2) The experimental set-up for the additional datasets seems to be unrealistic as hyperparameters for instance segmentation are derived from a grid search and it is unclear how a new user could find good parameters in the plugin without having access to already annotated ground-truth data or an extensive knowledge of the underlying implementations.

We agree that of course with any self-supervised method the user will need a sense of what a good outcome looks like; that is why we provide Google Colab Notebooks

(https://github.com/AdaptiveMotorControlLab/CellSeg3D/tree/main/notebooks) and the napari-plugin GUI for extensive visualization and even the ability to manually correct small subsets of the data and refine the WNet3D model.

We attempted to make this more clear with a new Figure 2 and additional functionality directly into the plugin (such as the grid search). But, we believe this “trade-off” for SSL approaches over very labor intensive 3D labeling is often worth it; annotators are also biased so extensive checking of any GT data is equally required.

We also added the “grid search” functionality in the GUI (please `pip install --upgrade napari-cellseg3d`; the latest v0.2.1) to supplement the previously shared Notebook (https://github.com/C-Achard/cellseg3d-figures/blob/main/thresholds_opti/find_best_threshold s.ipynb) and added a new YouTube video: https://www.youtube.com/watch?v=xYbYqL1KDYE.

(3) Obtaining segmentation results of similar quality as reported in the experiments within the napari plugin was not possible for me. I tried this on the "MouseSkull" dataset that was also used for the additional results in the paper.

Again we are sorry this did not work for you, but we added new functionality in the GUI and made a demo video (https://www.youtube.com/watch?v=U2a9IbiO7nE) where you either update your CellSeg3D code or watch the video to see how we obtained these results.

Here, I could not find settings in the "Utilities->Convert to instance labels" widget that yielded good segmentation quality and it is unclear to me how a new user could find good parameter settings. In more detail, I cannot use the "Voronoi-Otsu" method due to installation issues that are prohibitive for a non expert user and the "Watershed" segmentation method yields a strong oversegmentation.

Sorry to hear of the installation issue with Voronoi-Otsu; we updated the documentation and the GUI to hopefully make this easier to install. While we do not claim this code is for beginners, we do aim to be a welcoming community, thus we provide support on GitHub, extensive docs, videos, the GUI, and Google Colab Notebooks to help users get started.

Comments on revised versionMany of my comments were addressed well:- It is now clear that the results are reproducible as they are well documented in the provided notebooks, which are now much more prominently referenced in the text.

Thanks!

- My concerns about an unfair evaluation compared to CellPose and StarDist were addressed. It is now clear that the experiments on the mesoSPIM dataset are extensive and give an adequate comparison of the methods.

Thank you; to note we additionally added benchmarking of Cellpose and StarDist on the three additional datasets (for R1), but hopefully this serves to also increase your confidence in our approach.

- Several other minor points like reporting of the evaluation metric are addressed.I have changed my assessment of the experimental evidence to incomplete/**solid** and updated the review accordingly. Note that some of my main concerns with the usability of the method for segmentation tasks with more complex morphology / more crowded cells and with the napari plugin still persist. The main points are (also mentioned in Weaknesses, but here with reference to the rebuttal letter):- Method comparison on datasets with more complex morphology etc. are missing. I disagree that it is enough to do this on one dataset for a good method comparison.

We benchmarked WNet3D (our contribution) on four datasets, and to aid the readers we additionally now added Cellpose and StarDist benchmarking on all four. WNet3D performs favorably, even on the crowded and complex Mouse Skull data. See the new Figure 3 as well as the associated video: https://www.youtube.com/watch?v=U2a9IbiO7nE&t=1s.

- The current presentation still implies that CellSeg3d **and the napari plugin** work well for a dataset with complex nucleus morphology like the Mouse Skull dataset. But I could not get this to work with the napari plugin, see next points.- First, deriving hyperparameters via grid search may lead to over-optimistic evaluation results. How would a user find these parameters without having access to ground-truth? Did you do any experiments on the robustness of the parameters?- In my own experiments I could not do this with the plugin. I tried this again, but ran into the same problems as last time: pyClesperanto does not work for me. The solution you link requires updating openCL drivers and the accepted solution in the forum post is "switch to a different workstation".

We apologize for the confusion here; the accepted solution (not accepted by us) was *user specific* as they switched work stations and it worked, so that was their solution. Other comments actually solved the issue as well. For ease this package can be installed on Google Colab (here is the link from our repo for ease: https://colab.research.google.com/github/AdaptiveMotorControlLab/CellSeg3d/blob/main/not ebooks/Colab_inference_demo.ipynb) where pyClesperanto can be installed via: !pip install pyclesperanto-prototype without issue on Google Colab.

This (a) goes beyond the time I can invest for a review and (b) is unrealistic to expect computationally inexperienced users to manage. Then I tried with the "watershed" segmentation, but this yields a strong oversegmentation no matter what I try, which is consistent with the predictions that look like a slightly denoised version of the input images and not like a proper foreground-background segmentation. With respect to the video you provide: I would like to see how a user can do this in the plugin without having a prior knowledge on good parameters or just pasting code, which is again not what you would expect a computationally unexperienced user to do.

We agree with the reviewer that the user needs domain knowledge, but we never claim our method was for inexperienced users. Our main goal was to show a new computer vision method with self-supervised learning (WNet3D) that works on LSM and confocal data for cell nuclei. To this end, we made you a demo video to show how a user can visually perform a thresholding check https://www.youtube.com/watch?v=xYbYqL1KDYE&t=5s, and we added all of these new utilities to the GUI, thanks for the suggestion. Otherwise, the threshold can also be done in a Notebook (as previously noted).

I acknowledge that some of these points are addressed in the limitations, but the text still implies that it is possible to get good segmentation results for such segmentation problems: "we believe that our self-supervised semantic segmentation model could be applied to more challenging data as long as the above limitations are taken into account." From my point of view the evidence for this is still lacking and would need to be provided by addressing the points raised above for me to further raise the Incomplete/solid rating, especially showing how this can be done wit the napari plugin. As an alternative, I would also consider raising it if the claims are further reduced and acknowledge that the current version of the method is only a good method for well separated nuclei.

We hope our new benchmarking and clear demo on four datasets helps improve your confidence in our evidence in our approach. We also refined our over text and hope our contributions, the limitations and the advantages are now more clear.

I understand that this may be frustrating, but please put yourself in the role of a new reader of this work: the impression that is made is that this is a method that can solve 3D segmentation tasks in light-sheet microscopy with unsupervised learning. This would be a really big achievement! The wording in the limitation section sounds like strategic disclaimers that imply that it is still possible to do this, just that it wasn't tested enough.But, to the best of my assessment, the current version of the method only enables the more narrow case of well separated nuclei with a simple morphology. This is still a quite meaningful achievement, but more limited than the initial impression. So either the experimental evidence needs to be improved, including a demonstration how to achieve this in practice, including without deriving parameters via grid-search and in the plugin, or the claim needs to be meaningfully toned down.

Thanks for raising this point; we do think that WNet3D and the associated CellSeg3D package - aimed to continue to integrate state of the art models, is a non-trivial step forward. Have we completely solved the problem, certainly not, but given the limited 3D cell segmentation tools that exist, we hope this, coupled with our novel 3D dataset, pushes the field forward. We don’t show it works on the narrow well-separated use case, but rather show this works even better than supervised models on the very challenging benchmark Mouse Skull. Given we now show evidence that we outperform or match supervised algorithms with an unsupervised approach, we respectfully do think this is a noteworthy achievement. Thank you for your time in assessing our work.